# Basal MET phosphorylation is an indicator of hepatocyte dysregulation in liver disease

Sebastian Burbano de Lara [1,2,11], Svenja Kemmer [2,3,4,11], Ina Biermayer[1,2,11], Svenja Feiler[1,5], Artyom Vlasov[1], Lorenza A D'Alessandro[1], Barbara Helm[1], Christina Mölders[1,2], Yannik Dieter [1], Ahmed Ghallab [6,7], Jan G Hengstler[2,6], Christiane Körner [2,8], Madlen Matz-Soja[2,8], Christina Götz [2,9], Georg Damm[2,9], Katrin Hoffmann[2,5], Daniel Seehofer[2,9], Thomas Berg[2,8], Marcel Schilling [1✉], Jens Timmer [2,3,4,10✉] & Ursula Klingmüller [1,2✉]

## Abstract

**Chronic liver diseases are worldwide on the rise. Due to the rapidly increasing incidence, in particular in Western countries, metabolic dysfunction-associated steatotic liver disease (MASLD) is gaining importance as the disease can develop into hepatocellular carcinoma. Lipid accumulation in hepatocytes has been identified as the characteristic structural change in MASLD development, but molecular mechanisms responsible for disease progression remained unresolved. Here, we uncover in primary hepatocytes from a preclinical model fed with a Western diet (WD) an increased basal MET phosphorylation and a strong downregulation of the PI3K-AKT pathway. Dynamic pathway modeling of hepatocyte growth factor (HGF) signal transduction combined with global proteomics identifies that an elevated basal MET phosphorylation rate is the main driver of altered signaling leading to increased proliferation of WD-hepatocytes. Model-adaptation to patient-derived hepatocytes reveal patient-specific variability in basal MET phosphorylation, which correlates with patient outcome after liver surgery. Thus, dysregulated basal MET phosphorylation could be an indicator for the health status of the liver and thereby inform on the risk of a patient to suffer from liver failure after surgery.**

**Keywords** Western Diet; Dynamic Pathway Modeling; Hepatocyte Dysregulation; Fatty Liver Disease; HGF Signal Transduction
**Subject Categories** Molecular Biology of Disease; Proteomics; Signal Transduction

## Introduction

Chronic liver disease is a frequent pathology with increasing mortality rates. Hepatocytes, the most abundant cell type in the liver, play a central role in metabolism and for example store and degrade glycogen to ensure a constant supply of glucose in the blood. High caloric intake can disrupt this process and result in fatty liver disease (Riazi et al, 2022), a metabolic disorder characterized by the accumulation of lipid droplets in hepatocytes. If no secondary causes such as alcohol abuse or medication are identified, steatotic liver disease is categorized as metabolic dysfunction-associated steatotic liver disease (MASLD). In the past decades the incidence of MASLD has steadily increased. If sustained for a long period of time, MASLD can develop into metabolic dysfunction-associated steatotic steatohepatitis (MASH), fibrosis, cirrhosis and even hepatocellular carcinoma (Huang et al, 2021). These developments highlight that an understanding of the underlying mechanisms of disease development and progression is pivotal. Metabolic changes in different liver disease stages have been extensively characterized (Jia et al, 2014; Puri et al, 2007), whereas alterations in information processing of hepatocytes driven by high caloric diets have not yet been investigated in depth. Previous observations (Oe et al, 2005; Paranjpe et al, 2016; Tekkesin et al, 2011) suggest that hepatocyte growth factor (HGF) might play a central role in hepatic pathologies. HGF binds to the receptor tyrosine kinase MET on hepatocytes and triggers the activation of proliferative signal transduction by the mitogen activated kinase pathway (MAPK) and the phosphoinositide 3 kinase (PI3K)/AKT pathway. Through the negative feedback loop between S6K (MAPK) and IRS1 (PI3K/AKT), MET stimulation can affect the metabolic functions of the liver during regeneration (Hall et al, 2021). Furthermore, we utilized dynamic pathway modeling to disentangle the cross talk of the MAPK and PI3K/AKT pathways in these cells (D'Alessandro et al, 2015) and showed

[1]Division Systems Biology of Signal Transduction, German Cancer Research Center (DKFZ), Heidelberg, Germany. [2]Liver Systems Medicine against Cancer (LiSyM-Krebs), Heidelberg, Germany. [3]Institute of Physics, University of Freiburg, Freiburg, Germany. [4]FDM - Freiburg Center for Data Analysis and Modeling, University of Freiburg, Freiburg, Germany. [5]Department of General, Visceral and Transplant Surgery, Heidelberg University, Heidelberg, Germany. [6]Systems Toxicology, Leibniz Research Center for Working Environment and Human Factors, Technical University Dortmund, Dortmund, Germany. [7]Department of Forensic Medicine and Toxicology, Faculty of Veterinary Medicine, South Valley University, Qena 83523, Egypt. [8]Division of Hepatology, Clinic of Oncology, Gastroenterology, Hepatology, and Pneumology, University Hospital Leipzig, 04103 Leipzig, Germany. [9]Department of Hepatobiliary Surgery and Visceral Transplantation, University Hospital Leipzig, Leipzig University, 04103 Leipzig, Germany. [10]Signalling Research Centres BIOSS and CIBSS, University of Freiburg, Freiburg, Germany. [11]These authors contributed equally: Sebastian Burbano de Lara, Svenja Kemmer, Ina Biermayer.
✉E-mail: m.schilling@dkfz.de; jeti@fdm.uni-freiburg.de; u.klingmueller@dkfz.de

that both, ERK phosphorylation and PI3K activation, are required for proliferation of primary mouse hepatocytes (Mueller et al, 2015). As most liver pathologies are driven by increasing damage of the liver, consideration of HGF-induced signal transduction in hepatocytes, which is essential for liver regeneration, could be informative to predict patient outcome upon liver surgery. Currently, only postoperative descriptive scores such as the Clavien Dindo score (Dindo et al, 2004) or the comprehensive complication index (Slankamenac et al, 2013) are documented and so far no correlations with preoperative metrices could be established. The implementation of complex patient datasets for clinical decisions remains challenging and requires the development of comprehensive tools. A suitable approach could be the use of mechanism-based dynamic pathway models to integrate and exploit complex datasets (D'Alessandro et al, 2022; Kok et al, 2020; Oppelt et al, 2018) in order to guide clinical decisions.

In this work, we utilized the Western diet (WD) mouse as a preclinical model to study alterations in HGF-induced signal transduction occurring in liver disease. Data generated from primary murine hepatocytes of healthy and WD mice and from patient-derived primary human hepatocytes were used to calibrate a dynamic pathway model of HGF-induced signal transduction, which allowed us to resolve the molecular mechanism resulting in reduced AKT phosphorylation in WD hepatocytes. A patient-adapted mathematical model correlated the basal MET phosphorylation with patient outcome after liver surgery and thus suggests MET phosphorylation as an indicator for liver disease burden.

## Results

### Characterization of proteomic alterations in Western Diet (WD) primary hepatocytes identifies dysregulated pathways

The development of metabolic dysfunction-associated steatotic liver disease (MASLD) is characterized by the gradual accumulation of lipid droplets in hepatocytes. We hypothesized that these major structural changes have an impact on information processing and metabolic regulation in these cells. To induce a fatty liver-like phenotype, 8 weeks old C57BL/6N mice were fed with a high sugar high fat Western diet (WD) for up to 13 weeks. Age-matched mice fed with a standard diet (SD) served as controls. As expected, WD mice showed with a median weight of 39.4 g a significant increase in body weight, while SD mice remained at a median body weight of 29.7 g, (Fig. 1A). Bright-field microscopy of primary mouse hepatocytes isolated from these mice (Fig. 1B) revealed that lipid accumulation characterized by the formation of lipid droplets was indeed evident in the primary hepatocytes of the WD mice, but absent in those of the SD mice (Fig. 1C, black arrows). To characterize the diet-induced changes in the protein composition of steatotic hepatocytes, we analyzed the proteome of both SD and WD primary hepatocytes by mass spectrometry ($N = 9$ mice per condition) employing data independent acquisition (DIA). In total 4317 proteins were identified and a multidimensional scaling analysis provided evidence for major differences in the respective proteomes (Fig. 1D). Data analysis utilizing *limma* (Ritchie et al, 2015), identified 301 proteins as

differentially upregulated and 255 proteins as differentially downregulated in WD primary hepatocytes. To determine which pathways are primarily affected by feeding the WD, we performed an Ingenuity pathway analysis (Krämer et al, 2014) using significantly changed proteins as input (Benjamini and Hochberg adjusted $p$ value < 0.05 and log$_2$ fold change $<-0.5$ or >0.5). In Fig. 1F, the enriched categories of canonical pathways with top 10 highest and lowest $z$-score are displayed. In line with previous studies (Greco et al, 2008; Puri et al, 2007) the majority of the identified pathways were connected to metabolism, including glucose, fatty acids and cholesterol metabolism. However, surprisingly the top most upregulated pathway was kinetochore metaphase signaling. The kinetochore is a complex of proteins responsible for anchoring the spindle fibers to duplicating chromatids and pulling sister chromatids during the metaphase of the cell cycle apart. Detected members of the kinetochore metaphase signaling pathways (Fig. 1E) upregulated in WD primary hepatocytes were Macroh2a, H2ac20, H2az1 and H2ax, all linked to the nucleosome, and Zwint and Zw10, linked to kinetochore and mitosis, while Ppp1r10, a protein of the PTW/PP1 phosphatase complex controlling progression from mitosis to interphase, was downregulated. The HIF1α signaling pathway ranked second. The HIF1 α transcription factor has been linked in the context of the liver to lipid metabolism and vascular regulation during liver regeneration (Seo et al, 2020) and to enhanced proliferation of liver cancer cell lines (Tajima et al, 2009). Interestingly, the third most upregulated pathway in WD primary hepatocytes was ERK/MAPK signal transduction, a key pathway controlling proliferative responses. In sum, these proteomic alterations indicate that in steatotic hepatocytes the intricate network of metabolism and signal transduction controlling cell proliferation is dysregulated.

### Alterations in basal MET phosphorylation and AKT phosphorylation are characteristic features of WD primary hepatocytes

Hepatocytes in the liver are usually in a quiescent state and rarely proliferate in the absence of growth factors (Bottaro et al, 1991). The key growth factor controlling hepatocyte proliferation is HGF that binds to the receptor tyrosine kinase MET and triggers phosphorylation of Tyr 1232 and Tyr 1233 on the murine MET receptor (Tyr 1234 and Tyr 1235 on human MET). This enables the activation of PI3 kinase and MAP kinase signal transduction. To examine whether the uncovered proteomic alterations give rise to changes in the dynamics of HGF-induced signal transduction in WD primary hepatocytes, we designed based on our previous knowledge of the cross-talk of MAPK- and PI3K/AKT-pathways (D'Alessandro et al, 2015) and the link to proliferation (Mueller et al, 2015), dose- and time-resolved experiments to delineate by quantitative immunoblotting the differences in HGF-induced responses in WD and SD primary hepatocytes (Fig. 2A). The HGF dose response experiments showed that in SD hepatocytes the HGF-induced phosphorylation of MET (double phosphorylation on Tyr 1232 and Tyr 1233) increased with higher doses and reached saturation at 40 ng/ml HGF (Fig. 2B). Accordingly, the peak of HGF-induced phosphorylation of ERK (double phosphorylation on Thr 202 and Tyr 204 of ERK1/MAPK3 and on Thr 185 and Tyr 187 of ERK2/MAPK1) and of AKT (phosphorylation on

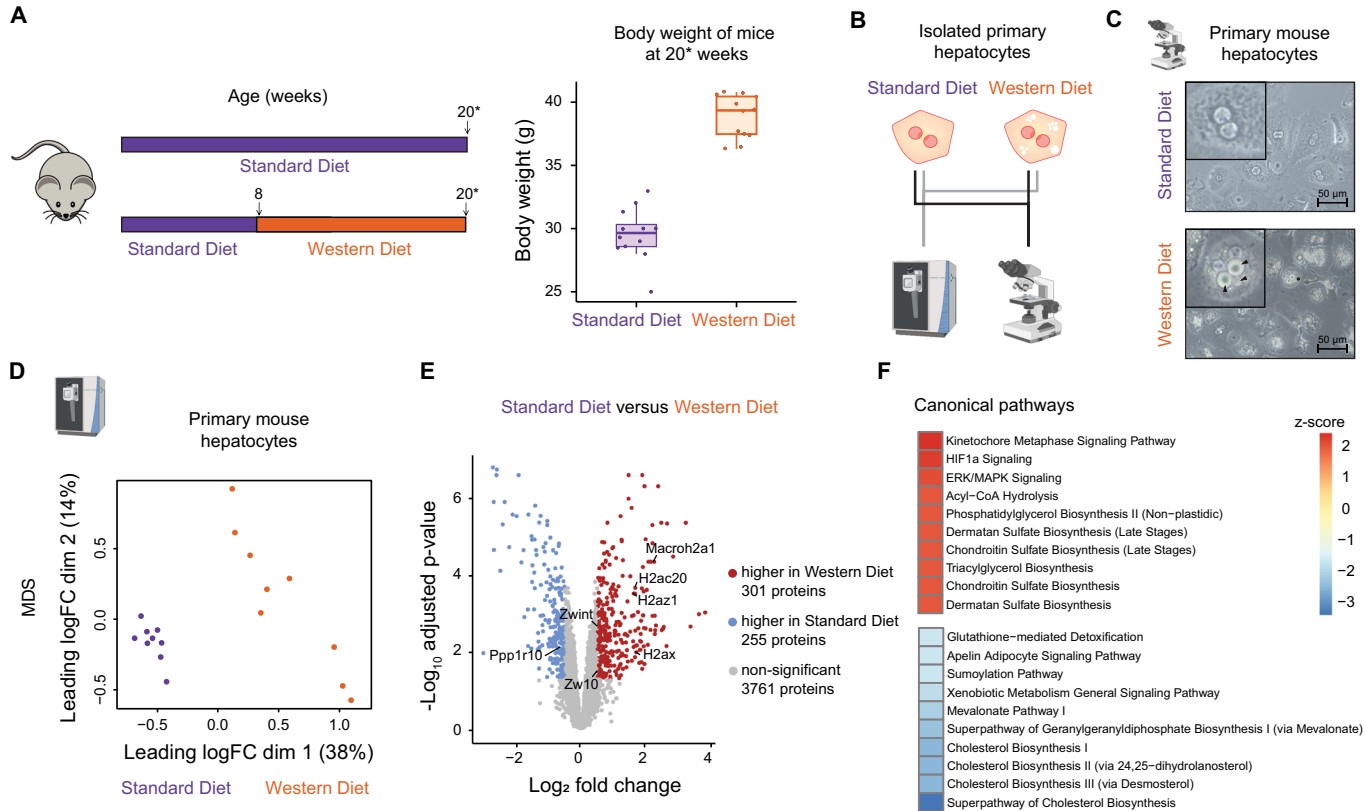

**Figure 1. Western diet-induced proteome alterations.**

(A) Schematic representation of the experimental setup. Mice were fed with standard diet (SD) for eight weeks and then switched to Western diet (WD) for 12–13 weeks or continued to receive SD as control. The body weight of SD mice is shown as boxplot with the center line indicating median (29.65 g); box limits indicate 25th (28.58 g) to 75th percentile (30.33 g). The lower and upper whiskers extend from the hinge to the smallest (28.0 g) or largest value (32.0) at most 1.5× interquartile range of the hinge. The body weight of WD mice is shown as boxplot with the center line indicating median (39.35 g); box limits indicate 25th (37.48 g) to 75th percentile (40.45 g). The lower and upper whiskers extend from the hinge to the smallest (36.3 g) or largest value (40.8) at most 1.5× interquartile range of the hinge. The dots represent data of single mice ($n = 9$ per diet). 20*: mice have an age of 20 or 21 weeks. (B) Primary mouse hepatocytes from SD and WD mice were isolated by liver perfusion, cultivated and characterized employing mass spectrometry and bright field imaging. (C) Exemplary bright field images from isolated primary mouse hepatocytes depict lipid droplet formation in hepatocytes derived from WD-fed mice, but not SD-fed mice. Arrows point to lipid droplets. (D) Multidimensional scaling analysis of the mass spectrometry-based hepatocyte proteome derived from SD and WD mice. Each dot represents the sample from one mouse ($n = 9$ per diet). (E) Up- and downregulated proteins in primary mouse hepatocytes derived from SD and WD mice were identified using the limma package by log fold change calculation and analysis of the adjusted $p$ value (limma-voom (Law et al, 2014)) as depicted in the volcano plot. Proteins with fold change <−0.5 and adjusted $p$ value <0.05, describing a downregulation in WD, are indicated in blue, proteins with fold change >0.5 and adjusted $p$ value < 0.05, representing an upregulation in WD, are indicated in red ($n = 9$ per diet). (F) The top 10 up- and downregulated pathways in WD primary mouse hepatocytes in comparison to SD primary mouse hepatocytes are depicted as identified by Ingenuity pathway analysis (Krämer et al, 2014). Source data are available online for this figure.

Thr 308, referred as pAKT Thr 308 and phosphorylation in both Thr 308 and Ser 473, referred as ppAKT Ser 473 as the employed antibody detects exclusively the second phosphorylation of AKT) reached saturation at 40 ng/ml HGF. In WD primary hepatocytes at high HGF doses, a similar extent of MET and ERK phosphorylation was observed. In contrast, the phosphorylation at basal level and the phosphorylation of MET upon low doses of HGF were higher in WD primary hepatocytes when compared to SD primary hepatocytes. Even more strikingly, at all HGF doses higher than 1 ng/ml, AKT phosphorylation was significantly reduced in WD primary hepatocytes. The comparison of the dynamics of HGF-induced signal transduction in SD and WD primary hepatocytes (Fig. 2C) revealed that the peak amplitude of the tyrosine kinase receptor MET was higher in SD primary hepatocytes, while the basal amount of phosphorylated MET was elevated in WD primary hepatocytes. In line with the notion that ligand mediated activation of MET induces its internalization and degradation (Jeffers et al, 1997), we observed a decline of total MET after HGF stimulation in SD primary hepatocytes, but this decline was reduced in WD hepatocytes. As expected, the phosphorylation of ERK correlated in SD and WD primary hepatocytes with the respective phosphorylation dynamics of MET. Interestingly, although the total amount of AKT was comparable between SD and WD primary hepatocytes, in line with the results obtained in the dose response experiment (Fig. 2B), HGF-induced phosphorylation of AKT was much reduced in WD primary hepatocytes compared to those from SD mice (Fig. 2C). In conclusion, our results suggested that exposure of hepatocytes to a high sugar and high fat diet results in a dysregulation of HGF-induced proliferative signal transduction.

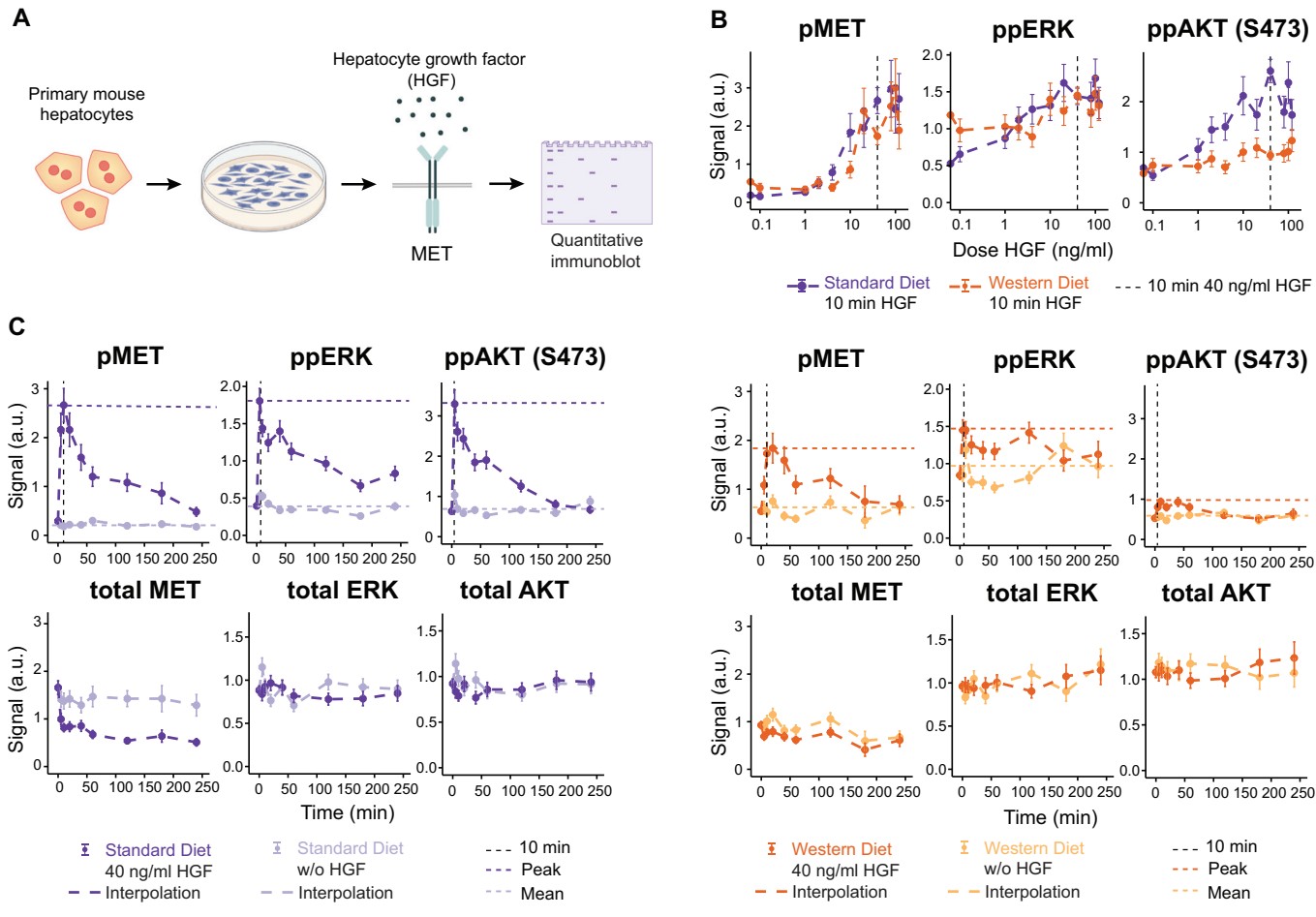

**Figure 2. HGF-induced activation of signal transduction in SD and WD hepatocytes.**

(A) Experimental design of HGF stimulation experiments in primary mouse hepatocytes. Isolated cells were seeded and stimulated with hepatocyte growth factor (HGF). HGF-induced signal transduction was analyzed by quantitative immunoblotting. (B) HGF dose dependency of MET, ERK and AKT phosphorylation in SD and WD primary mouse hepatocytes. Cells were stimulated with indicated doses of HGF and phosphorylation of MET, ERK and AKT was quantified by immunoblotting after 10 min. Signal is shown in arbitrary units (a.u.). Data points are displayed as dots with error bars representing 1σ confidence interval estimated from biological replicates ($n = 3$–9 per diet and dose) using a combined scaling and error model. Dashed curves represent linear interpolations. The dose of 40 ng/ml HGF is indicated as a vertical dashed line. (C) Time course measurements of HGF-induced signal transduction in primary mouse hepatocytes of SD and WD mice. Cells were stimulated with 40 ng/ml HGF for up to 4 h and the phosphorylation as well as the abundance of MET, ERK and AKT was quantified by immunoblotting. Signal is shown in arbitrary units (a.u.). Data points are displayed as dots with error bars representing 1σ confidence interval estimated from biological replicates ($n = 3$–9 per diet and time point) using a combined scaling and error model. Dashed curves represent linear interpolations. Horizontal dashed lines indicate basal and peak signal levels. The 10 min time point is indicated by a vertical dashed line. Source data are available online for this figure.

## Dynamic pathway model-based analysis identifies basal MET phosphorylation and protein abundances as dysregulated in primary mouse hepatocytes

To unravel the molecular mechanism leading to decreased HGF-induced AKT phosphorylation in WD primary hepatocytes, we developed a mechanistic mathematical model of HGF-induced signal transduction including the nutrient sensor mTOR that modulates MAPK and PI3K/AKT signal transduction as a function of the available metabolites. We hypothesized that the high sugar and high fat diet affects mTOR signal transduction and as a consequence impacts the activation of the pro-mitogenic ribosomal protein S6 as well as the phosphorylation of ATK (Fig. 3A).

The dynamic pathway model is based on coupled ordinary differential equations (ODEs) and describes the interconnection between HGF-induced signal transduction and mTOR. In the mathematical model, MET is subject to constant production and degradation, while phosphorylated MET is degraded with a different rate. The signal emanating from the activated MET receptor is propagated by two distinct signal transduction pathways, MAPK and PI3K/AKT. In the MAPK cascade, phosphorylated MET leads to MEK phosphorylation, which in turn phosphorylates ERK. In the PI3K pathway, phosphorylated MET recruits PI3K, which induces the first phosphorylation of AKT, at Thr 308 (pAKT). Both phosphorylated ERK and AKT phosphorylated on Thr 308 can inactivate the TSC complex by phosphorylating TSC2, which allows mTORC1 formation and

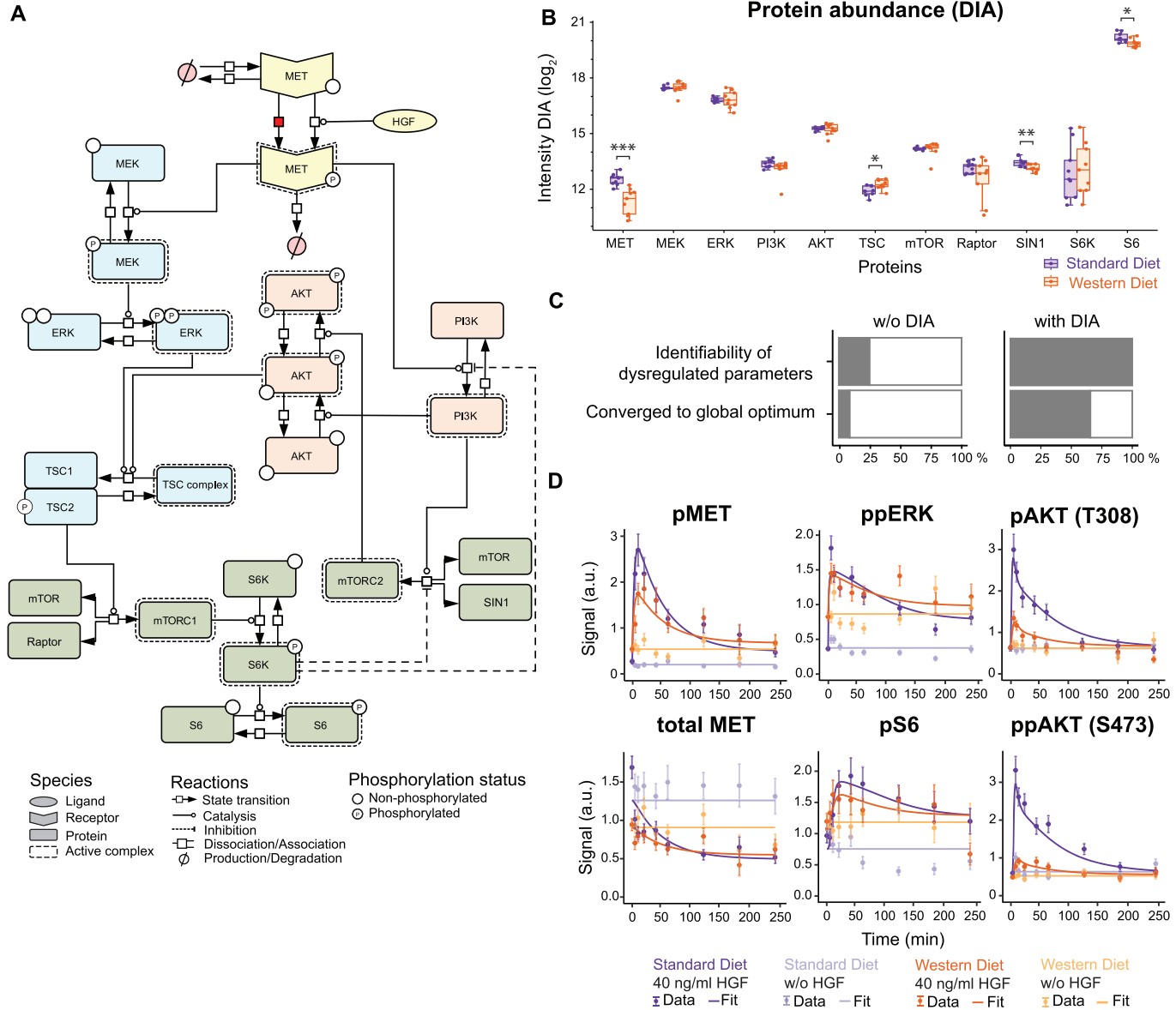

**Figure 3. Modeling WD-induced alterations in HGF signal transduction.**

(A) The structure of the mathematical model capturing HGF-induced signal transduction via the MAPK cascade (blue), the PI3K pathway (red) and mTOR signaling (green) is displayed according to Systems Biology Graphical Notation (Le Novere et al, 2009). All parameter values were implemented as identical for WD and SD hepatocytes except for the basal MET phosphorylation rate, indicated by the red box, and protein abundances. (B) Measurements of protein abundances derived from primary mouse hepatocytes. Lysates of unstimulated hepatocytes were subjected to data-independent mass spectrometry analysis. Resulting data was LFQ normalized and represented as boxplot: center line indicates median; box limits indicate 25th to 75th percentiles. The lower and upper whiskers extend from the hinge to the smallest or largest value at most 1.5× interquartile range of the hinge. Dots represent data of single mice (n = 9 per diet). p values were calculated using two-tailed t test (MET ***0.0002, TSC *0.01, SIN1 **0.006, S6 *0.01). (C) Impact of the DIA data on parameter identifiability and convergence of the mathematical model. The identifiability of the twelve dysregulated parameters increases from 25% to 100% upon DIA data incorporation, while the convergence to the global optimum during optimization increases from 9% to 66%. (D) Model calibration with time-resolved immunoblot measurements for MET, ERK, S6 and AKT phosphorylation and MET abundance upon stimulation with 40 ng/ml HGF. Data points are displayed as dots along with error bars representing 1σ confidence interval estimated from biological replicates (n = 3–9 per diet and time point) using a combined scaling and error model. Model trajectories are depicted as solid lines.

activation. Active mTORC1 phosphorylates S6K that in turn activates the pro-mitogenic ribosomal protein S6. Additionally, activated PI3K induces mTORC2 complex formation, which leads to phosphorylation of AKT on the second phosphorylation site Ser 473 resulting in AKT phosphorylated on Thr 308 and Ser 473 (ppAKT) and serving as a readout for the activation of the

mTORC2 complex. Two negative feedback loops originating from phosphorylated S6K are represented in the mathematical model in a condensed manner (Fig. 3A, dashed lines). First, activated S6K leads to the inhibition of the mTORC2 complex. Second, activated S6K phosphorylates and inhibits IRS1, which prevents activation of PI3K. In sum, our mathematical model includes 26

ordinary differential equations (for model reactions and observation functions, see Dataset EV1 and Dataset EV2) and 23 different model states.

To capture the dynamic properties of the system, the parameters of the mathematical model were calibrated based on the quantitative dose- and time-resolved immunoblot data shown in Fig. 2. Additionally, we generated data on the HGF dose-dependent phosphorylation of AKT on the phosphorylation site Thr 308 (pAKT) as well as on the amount of total MET, total AKT and total ERK (Fig. EV1A, data points). Further, the time-resolved dynamics of AKT phosphorylation on Thr 308 and of pS6 (Fig. 3D, data points) and of total S6 (Fig. EV1B, data points) were recorded. In total, 491 data points generated under 22 experimental conditions were employed for the calibration of our mathematical model. Since previous model-based investigations revealed that differences in the basal protein abundance greatly affect the dynamics of information processing (Adlung et al, 2017), we assumed as a start that all basal protein levels were different in SD and WD primary hepatocytes, and thus were considered as dysregulated parameters. To investigate if, in addition, dynamic parameters such as phosphorylation rates were affected by the Western diet, we performed a comparison of different model hypotheses by evaluating the resulting model fits based on the Bayesian information criterion (BIC) (Fig. EV1C). Interestingly, our analysis revealed that all but one dynamic parameter could be assumed as identical retaining a good fit to the data of SD and WD primary hepatocytes. For the accurate model-based representation of the data for both conditions it is necessary and sufficient if only the dynamic parameter for the basal (HGF-independent) phosphorylation rate of MET is increased in WD primary hepatocytes (Fig. 3A, reaction marked in red). To evaluate, whether the abundance of total MET could serve as an indicator for the levels of phosphorylated MET, we examined their correlation. The results displayed in Fig. EV1D showed that differences in the abundance of total MET do not allow to conclude on the levels of basal phosphorylated MET.

Thus, twelve dysregulated parameters appear to be required to describe the altered HGF-induced signal transduction in primary mouse hepatocytes induced by the Western diet, comprising the basal phosphorylation rate of MET and differences in the abundance of eleven proteins. To assess the identifiability of these twelve dysregulated parameters, we calculated profile likelihood-based confidence intervals (Raue et al, 2009) for each parameter (Fig. EV2A). This analysis showed that only three of these parameters where identifiable for both conditions, i.e., had defined confidence intervals (Fig. 3C). Based on previous experience (Adlung et al, 2017), we reasoned that inclusion of the basal abundance of all key protein species would improve parameter identifiability. Therefore, total protein lysates of SD and WD primary hepatocytes were examined by global proteomics employing mass spectrometric analysis operated in the data independent acquisition (DIA) mode that facilitated reproducible coverage and reliable quantification. These determinations revealed a significant decrease in the intensity of MET, SIN1 and S6 and a significant increase of TSC in the WD primary hepatocytes, while the intensity of the other protein species was comparable between SD and WD primary hepatocytes, respectively (Fig. 3B). Since the relative amount, which is based on the intensity determined by global proteomics, scales with the concentration of the examined proteins, we exemplarily determined the abundance of AKT in the SD and WD primary hepatocytes with quantitative immunoblotting (Fig. EV3A). These quantifications and the knowledge of the average protein content of the primary hepatocytes allowed us to estimate the number of AKT molecules per cell, which was included as additional information in the mathematical model. By linking the relative amount of AKT protein determined as intensity by our mass spectrometry-based DIA measurements with the corresponding absolute amount of AKT molecules per cell determined by quantitative immunoblotting, the model could infer the corresponding protein concentrations for all proteins of interest. With this additional information the mathematical model was able to estimate all twelve dysregulated parameters with narrow confidence intervals in both conditions (Fig. EV2B). Importantly, the inclusion of the absolute values for the protein abundance of SD and WD primary hepatocytes increased the identifiability of the dysregulated parameters from 25% to 100% and the convergence to the global optimum from 9% to 66% (Figs. 3C and EV2C,D). As a result, the final mathematical model, which included only one diet-specific dynamic parameter, the basal MET phosphorylation rate, and 11 diet-specific parameters for protein abundance, was able to explain the increased basal phosphorylation and reduced maximal induction of pMET, pERK and pS6 as well as the reduced phosphorylation of AKT in response to HGF stimulation in WD hepatocytes (Fig. 3D). Further, the model trajectories were in agreement with the experimentally observed HGF dose-dependent dynamics of pMET, pERK, pAKT (Thr 308) and ppAKT (Thr 308, Ser 473) in WD and SD primary hepatocytes (Fig. EV1A). The final mathematical model was capable of capturing the dynamics of total MET (Fig. 3D) as well as the differences in the total protein amounts of the analyzed proteins (Fig. EV1A,B) and was in line with the protein concentrations determined by mass spectrometry-based DIA measurements in SD and WD primary hepatocytes (Fig. EV3B).

The development of the mathematical model pointed to an important role of the mTOR pathway in regulating AKT phosphorylation in WD primary hepatocytes. Therefore, we utilized our global proteome data of SD and WD primary hepatocytes and tested if hallmark genes encoding proteins of mTORC1 signal transduction as listed in GSEA are differentially regulated in WD primary hepatocytes compared to SD hepatocytes. Of the 200 genes listed, 129 proteins were detected across all samples. The expression values were scaled per protein and compared between primary hepatocytes from 9 WD mice and 9 SD mice. As shown in Fig. EV3C, the clustering of the data showed that based on the hallmarks of mTORC1 signal transduction, SD and WD mice separate from each other, confirming our observation.

In sum, we established a mathematical model of HGF-signal transduction that was able to explain the alterations in HGF-induced MET and AKT phosphorylation dynamics in WD primary hepatocytes (for estimated parameter values, see Dataset EV3 and Dataset EV4). Importantly, the inclusion of the metabolite sensing pathway mTOR, in form of a negative feedback loop between mTORC1 and mTORC2, proved to be essential to gain insights into the molecular mechanisms causing the reduced AKT phosphorylation in WD hepatocytes: The increased basal MET phosphorylation in WD hepatocytes results in an increased basal

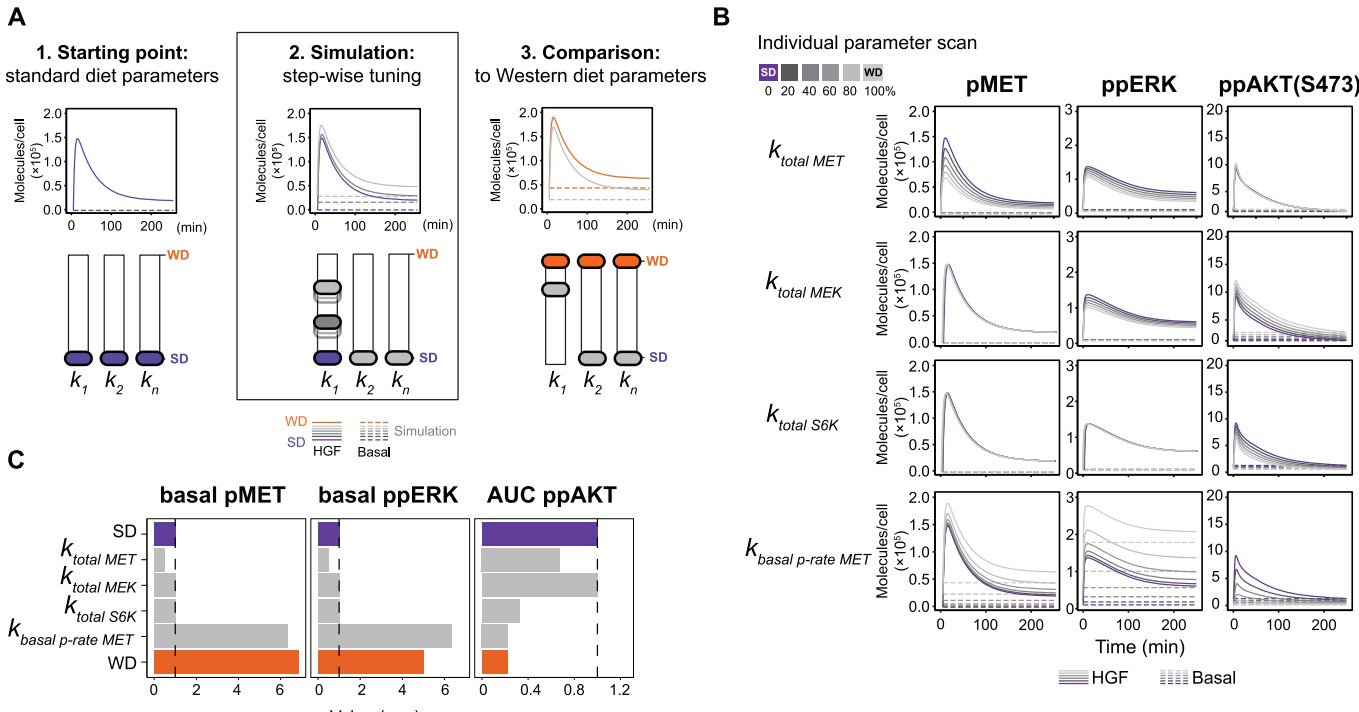

**Figure 4. Influence of dysregulated parameters on protein dynamics.**

(A) Schematic overview of the simulation analysis for diet-specific parameters. 1. All parameters were fixed to the estimates for SD hepatocytes. 2. Step-wise tuning of one dysregulated parameter at a time until it reached the value estimated for WD hepatocytes. 3. Model simulations were compared to the WD signaling dynamics.
(B) Individual parameter scan of one dysregulated parameter at a time ($k_{total\ MET}$, $k_{total\ MEK}$, $k_{total\ S6K}$, $k_{basal\ p-rate\ MET}$). The value for the indicated parameter was gradually shifted from the SD estimate (purple) to the WD estimate (light gray) as described in (A). The model simulations for the phosphorylation dynamics of MET, ERK and AKT are displayed in molecules/cell. Solid lines indicate model trajectories after HGF stimulation and dashed lines indicate basal levels. (C) Quantitative analysis of the ability of dysregulated parameters to reproduce WD-specific features (basal pMET, basal ppERK, AUC ppAKT). We reoptimized each dysregulated parameter individually in the range between SD and WD estimates to determine the best fit for the three features. Colored bars represent the feature value as determined from the original model fit for SD and WD. Gray bars indicate the optimized feature value for dysregulated parameters. Source data are available online for this figure.

phosphorylation of S6K via mTORC1, which in turn inhibits mTORC2 and PI3K activation and as a consequence, results in reduced AKT phosphorylation.

## The basal phosphorylation rate parameter of MET is sufficient to explain altered HGF-induced signal transduction in WD hepatocytes

To investigate the individual impact of the twelve diet-specific parameters on WD-specific changes in the dynamics of HGF-induced signal transduction, we utilized our mathematical model to perform a simulation analysis. As a starting point, all model parameters were set to the estimates for SD primary hepatocytes, which allowed to capture the dynamics of HGF signal transduction in these hepatocytes (Fig. 4A, 1. Starting point). Subsequently, we gradually (20% intervals) changed only one dysregulated parameter at a time until it reached the parameter value estimated for WD hepatocytes (Fig. 4A, 2. Simulation). We performed this analysis with all twelve parameters, which allowed us to assess the individual impact of each specific parameter by comparing the simulation results to the dynamics in WD primary hepatocytes (Fig. 4A, 3. Comparison). Since we identified the increased basal phosphorylation of MET and ERK and the decreased

area under the curve (AUC) of ppAKT upon HGF stimulation as main features differing between SD and WD primary hepatocytes, we simulated the trajectories of pMET, pERK and ppAKT (Thr 308, Ser 473). The model simulations (Figs. 4B and EV4) showed the largest effects when altering the basal protein abundance of MET ($k_{total\ MET}$), MEK ($k_{total\ MEK}$) and S6K ($k_{total\ S6K}$) as well as the basal phosphorylation rate of MET ($k_{basal\ p-rate\ MET}$). However, most of the parameters could not reproduce the features observed for WD primary hepatocytes. For example, a shift of $k_{total\ S6K}$ to the WD estimate resulted in a decrease in AKT phosphorylation but had no effect on the basal MET and ERK phosphorylation levels. Interestingly, when shifting the value of $k_{basal\ p-rate\ MET}$ to the estimated value for WD primary hepatocytes, all three features observed in WD primary hepatocytes were reproduced. Thus, only $k_{basal\ p-rate\ MET}$ was essential to reproduce all three features observed for WD primary hepatocytes.

To quantitatively asses how accurately these three features could be reproduced by the described parameter tuning, we performed an optimization analysis. During the analysis the mathematical model can choose an arbitrary value between the values in SD and WD for the analyzed parameter maximizing the reproducibility of the true WD features. In line with the results obtained by the simulation analysis, the only parameter that quantitatively reproduced the

WD-specific features was the basal MET phosphorylation rate (Fig. 4C), supporting our hypothesis of its key role as driver for the WD-specific alterations in HGF-induced signal transduction.

## WD primary hepatocytes show an increased proliferative behavior

As a consequence of the increased basal MET phosphorylation rate, we observed an elevated basal phosphorylation of the pro-mitogenic ribosomal protein S6 in WD primary hepatocytes. Therefore, we hypothesized that these changes might influence proliferative responses in the presence and absence of HGF. To explore this hypothesis, we fed 8 week old mice expressing the Fluorescent Ubiquitination-based Cell Cycle Indicator (Fucci2) (Abe et al, 2013) with either SD or WD for 12 weeks and isolated primary murine hepatocytes (Fig. 5A). For each condition, we monitored ten individual primary hepatocytes derived from three SD and three WD mice, respectively, by live cell microscopy and acquired time-resolved data on the changes in fluorescence intensities of the FUCCI system for up to 65 hours (Fig. 5B). To quantify the cell cycle entries as a measure for proliferative behavior, we performed single-cell tracking and counted cell cycle entries of each hepatocyte defined as the transition from G1/S to G2/M phase (Fig. 5C). For 13 out of 30 WD primary hepatocytes (43%) cell cycle entry was observed within the observation time, whereas only 7 out of 30 SD primary hepatocytes (23%) showed such a response. This observation indicates that already in the absence of HGF there is a marked increase in cell cycle entries in WD compared to SD primary hepatocytes. Upon HGF stimulation, all tracked SD and WD primary hepatocytes showed cell cycle progression, confirming that growth factor responsiveness was maintained in WD primary hepatocytes. Importantly, in response to HGF stimulation 28 out of 30 WD primary hepatocytes (94%) underwent two to four rounds of cell cycle progression within the observation time. In contrast, only 11 out of 30 SD primary hepatocytes (36%) showed two or three rounds of cell cycle progression. To corroborate these findings, we utilized the fluorescent dye SYBR Green, which binds to double-stranded DNA, to quantify the DNA content in SD and WD primary hepatocytes left unstimulated or stimulated with HGF. In agreement with the results obtained with the FUCCI mouse derived primary hepatocytes, WD primary hepatocytes showed an increase in DNA content even in the absence of HGF, indicating that these primary hepatocytes were able to proliferate independently of HGF stimulation (Fig. 5D). Taken together, these results revealed that proliferative responses are enhanced in WD primary hepatocytes even in the absence of HGF. These observations imply that the identified increase in basal MET phosphorylation in WD primary hepatocytes is an indicator of altered signal transduction enabling HGF-independent proliferation.

## Basal MET phosphorylation is indicative for liver disease burden in patient-derived primary hepatocytes

Our observation that basal MET acts as an integrator of structural and metabolic alterations during the progression of chronic liver disease in mice, let us to propose that basal MET phosphorylation could be a useful indicator for the burden of liver disease in humans. To test this hypothesis, we first examined in primary hepatocytes of three patients with 0%, 15% or 40% steatosis the levels of basal phosphorylated MET and total MET by quantitative immunoblotting (Fig. EV5). The quantification of the results showed that in relation to the levels of total MET there was indeed a significant increase in phosphorylated MET in the highly steatotic patient hepatocytes (Fig. EV5A), strikingly resembling the findings in the WD primary hepatocytes. Likewise, the phosphorylation of AKT after HGF stimulation was significantly reduced in the highly steatotic primary hepatocytes (Fig. EV5B) suggesting that the effects we uncovered in the preclinical mouse model also occurred in patients. To further dissect the underlying mechanism and the relation to clinical outcome, we analyzed HGF-induced signal transduction in patient-derived primary human hepatocytes isolated from tumor-free tissue of seven patients with different liver pathologies that underwent partial liver hepatectomy (see Dataset EV5 for patient anamnesis). The primary human hepatocytes were stimulated with HGF and cellular lysates taken at time points up to 120 min were analyzed by quantitative immunoblotting to generate time-resolved information on the dynamics of HGF-induced signal transduction in the hepatocytes of the individual patients (Fig. 6A) In addition, the proliferation behavior was also determined (Fig. EV6A). We examined the HGF-induced phosphorylation dynamics of MET (Tyr 1234 and Tyr 1235), ERK (Thr 202 and Tyr 204 on ERK1, Thr 185 and Tyr 187 on ERK2), AKT (Ser 473) (Fig. 6A, data points) and S6K (Thr 389) in the patient-derived hepatocytes (Fig. EV6B). Likewise, the total amounts of MET, ERK, AKT and S6K were determined by quantitative immunoblotting (Fig. EV6B) and complemented by protein abundances quantified by DIA-based mass spectrometry (Fig. EV6C, box plots). Our dynamic pathway model for HGF-induced signal transduction developed for murine SD and WD-derived primary hepatocytes was adapted to analyze the time-resolved data obtained for the patient-derived hepatocytes. The majority of the model parameters was kept as estimated for the murine system. The model parameters identified as dysregulated in the primary hepatocytes from WD mice, comprising protein abundances and the basal MET phosphorylation rate, were adapted to the conditions in the hepatocytes of individual patients. However, this model was yet insufficient to adequately describe the human data (98 parameters, BIC = 805). Therefore, we included in addition as human-specific parameters the HGF-induced MET phosphorylation rate as well as the degradation rate of the activated MET receptor (100 parameters, BIC = 564). These findings are supported by previous observations reporting a difference in the binding affinity of HGF to MET between mice and humans (Bussolino et al, 1992), which affects HGF binding to the receptor and as a consequence receptor phosphorylation and degradation of the ligand-receptor complex. In total 852 data points were used for the calibration of 100 human- and patient-specific parameters (see Dataset EV6 for estimated model parameters of primary human hepatocytes). The calibrated dynamic pathway model of HGF signal transduction in primary human hepatocytes was able to capture the basal (Figs. 6A and EV6B, dashed lines) as well as the HGF-induced patient-specific dynamics (Figs. 6A and EV6B, solid lines) of all pathway components. The patient data (displayed in log scale) was more heterogenous but overall resembled the dynamic behavior observed in the primary hepatocytes of the inbred mouse model. Further, the model fits for total protein abundances were in line with the global proteome measurements acquired by DIA-mass

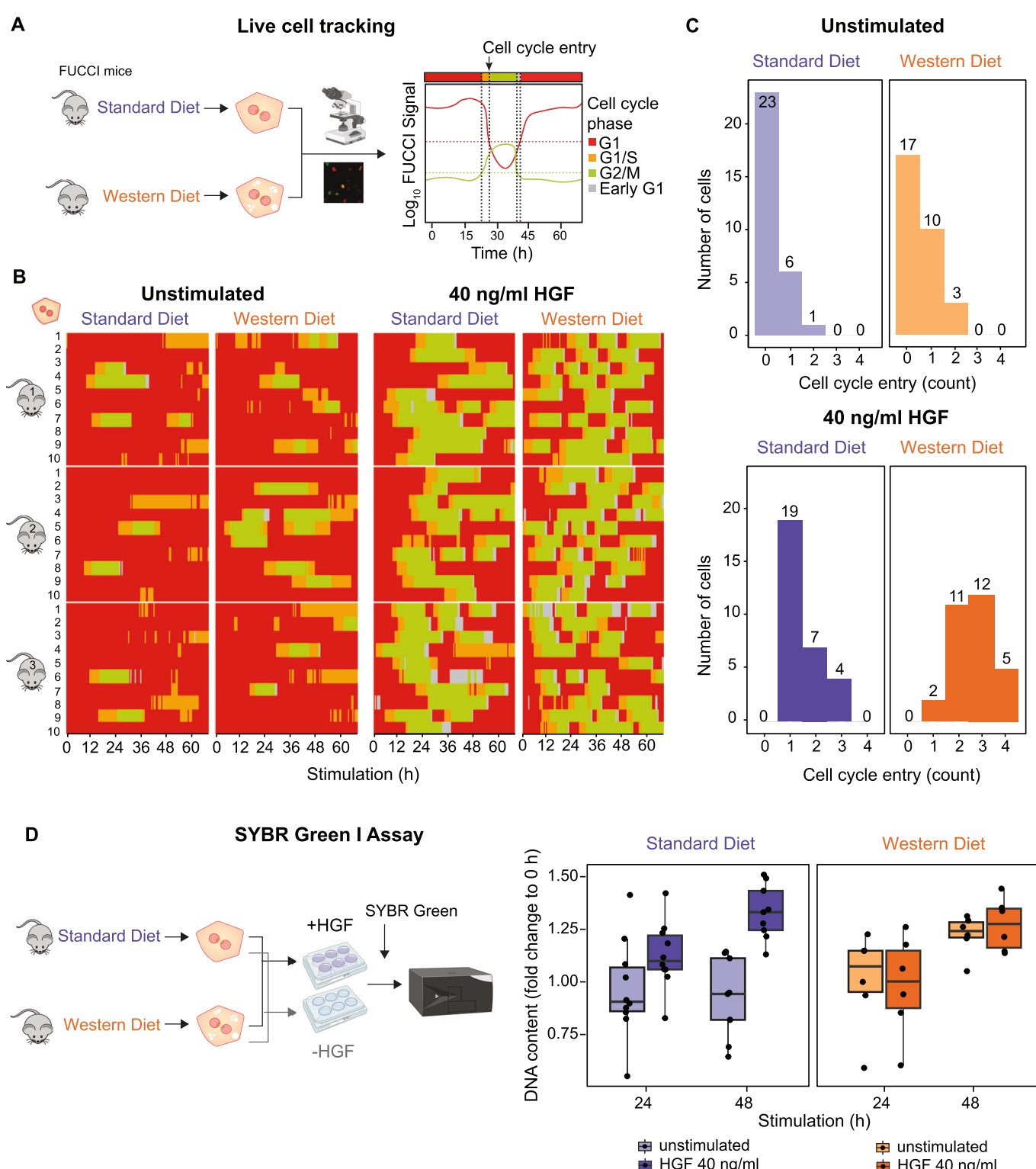

spectrometry (Fig. EV6C, points). Both experimental data and model trajectories revealed a patient-to-patient difference in the basal MET phosphorylation levels of untreated primary human hepatocytes (Fig. 6A). Since our model-based analyses of HGF signal transduction in hepatocytes of mice with WD-induced chronic liver disease suggested a relation between the basal MET

phosphorylation level and disease burden, we investigated the information encoded in the dynamics of HGF-induced signal transduction in patients. To this aim, we utilized our mathematical model calibrated for the patient-derived primary hepatocytes (Fig. 6A) to calculate the patient-specific basal MET phosphorylation rate, the AUC of ppAKT and the MET abundance. In addition,

**Figure 5. Altered proliferation of WD hepatocytes.**

(A) SD and WD mice carrying the Fucci2 cell cycle reporter were used to track cell cycle entries of primary mouse hepatocytes via live cell imaging. Cells were transduced with adeno-associated viral vectors encoding Histone2B–mCerulean. The FUCCI signal indicates the cell cycle phase of a cell at a given time and enables tracking of cell cycle entry. (B) Primary mouse hepatocytes of mice carrying the Fucci2 cell cycle reporter were stimulated with 40 ng/ml HGF or left untreated. Live cell microscopy of hepatocytes from SD and WD mice was performed with sampling rate of 15 min for up to 65 h. Ten exemplary hepatocytes of three SD and three WD mice each were tracked. The time course of cell cycle phases G1, G1/S, and G2/M and early G1 are displayed for each cell. (C) Quantification of cell cycle entries per cell. A cell cycle entry was considered if cells transited from S to G2-phase as indicated in (B). Shown are the number of cell cycle entries as a histogram indicating the number of cells that underwent a certain amount of cell cycle entries in each condition. (D) Primary mouse hepatocytes from SD and WD wildtype mice were stimulated with 40 ng/ml HGF or left untreated. Using SYBR Green I Assay, the DNA content of cells was measured at 0 h and after 24 h and 48 h. The DNA content as fold-change (FC) to 0 h is displayed as a boxplot: center line indicates median; box limits indicate 25th to 75th percentiles. The lower and upper whiskers extend from the hinge to the smallest or largest value at most 1.5× interquartile range of the hinge. Dots represent data of single mice ($n = 9$ for SD and $n = 6$ for WD). Source data are available online for this figure.

we introduced ratios of these characteristic features accounting for interconnected complexity. We also included the quantification of the HGF-induced proliferation of the patient-derived human hepatocytes as a proxy for the proliferation potential at the organ level. To test how informative the model-based characteristics and these calculated ratios and quantifications are, we correlated them to several patient-specific clinical parameters (Dataset EV7), which we divided in three subgroups: (i) Pre- and post-operative blood metrics: hepatocyte growth factor (HGF), Interleukin 6 (IL6), Interleukin 8 (IL8), Platelet-derived growth factor (PDGF); (ii) patient features, such as the age, the body mass index (BMI), the Fibrosis score and the Charlson Comorbidity Index (CCI); and (iii) the patient outcome after hepatectomy including intensive care and hospitalization days, the Clavien Dindo score and the complication index (Fig. 6B). Surprisingly, the ex-vivo proliferation of hepatocytes did not correlate with the patient outcome, emphasizing the complexity of the proliferative response in the liver. The AUC of ppAKT as well as the corresponding ratio $k_{basal\ p\text{-rate}\ MET}/AUC_{ppAKT}$ did not correlate with the patient outcome but to the post-operative concentrations of HGF and IL8. However, the basal MET phosphorylation rate, $k_{basal\ p\text{-rate}\ MET}$, showed a significant correlation with three of four patient outcome measures. This correlation was also present for the ratio $k_{basal\ p\text{-rate}\ MET}/k_{total\ MET}$, but to a smaller extend. Of the four blood metrix factors, HGF, IL6, IL8 and PDGF that were determined in longitudinal blood samples, only PDGF showed an anticorrelation with $k_{basal\ p\text{-rate}\ MET}$ at all time points. Lastly, we corrected for patient specific features by performing a partial correlation to remove effects of confounding factors. After correction for the confounding factors age, BMI, fibrosis and CCI, the correlation of patient outcome with proliferation and $k_{total\ MET}$ increased, albeit not to a significant extent. Importantly, the significant correlation between $k_{basal\ p\text{-rate}\ MET}$ and $_{basal\ p\text{-rate}\ MET}/k_{total\ MET}$ with complication index and hospitalization days was maintained. Taken together our results suggested that $k_{basal\ p\text{-rate}\ MET}$ and $k_{basal\ p\text{-rate}\ MET}/k_{total\ MET}$ might provide suitable measures to predict patient outcome.

As the parameter estimation of $k_{basal\ p\text{-rate}\ MET}$ cannot be easily introduced in a clinical setup, we tested if the basal phosphorylation of MET, which was used to parametrize the mathematical model, could describe the patient outcome. In Fig. 6C, the patients were sorted by the model-derived estimates of the basal MET phosphorylation rate regarding potential clinical outcome. The values of pMET that were experimentally determined by quantitative immunoblotting for each patient, corroborated with the order of the patients based on the basal MET phosphorylation rate. In line with the correlation analysis, Clavien Dindo score, complication index and especially hospitalization time directly correlated with both model-derived basal MET phosphorylation rate and experimentally measured basal pMET levels. In conclusion, our results identified the basal phosphorylation of MET as key node integrating alterations in hepatocytes and suggest it as an informative metric (Fig. 6D) for liver disease burden and patient recovery after liver surgery.

## Discussion

In this study, we characterize Western diet-induced phenotypic changes in murine hepatocytes and employ dynamic pathway modeling to resolve the underlying molecular mechanisms resulting in altered intracellular signal transduction and enhanced proliferation of WD hepatocytes. A key observation was that the basal MET phosphorylation is increased in WD mice after 12 weeks of WD-feeding, while conversely the amplitude of the MET phosphorylation dynamic is reduced. These insights required the generation of high-quality quantitative data on the dynamics of HGF signal transduction and thus shed new light on the complex regulation of MET in MASLD, while MET was so far primarily studied in the context of liver regeneration or cancer (Bottaro et al, 1991; Paranjpe et al, 2016). With our mathematical modeling approach, we resolve that the elevated levels of basal MET phosphorylation result in surprisingly strong inhibition of AKT phosphorylation in WD hepatocytes. Conversely, it was reported in the context of type II diabetes and fatty liver disease that increased phosphorylation of AKT was an indicator of insulin response and consequently a good prognostic marker for reduced steatosis (Vivero et al, 2021). In this context, it is relevant to mention that others have presented evidence for a heterodimeric interaction between insulin receptor and MET (Fafalios et al, 2011), which could be involved in the HGF independent basal phosphorylation of MET in WD mice. This heterodimer and the known interactions between MET and the insulin receptor substrate 1 and 2 (IRS1 and IRS2) (Fafalios et al, 2011) (DeAngelis et al, 2010) have not been explicitly included in our model, but could be integrated in the future to further elucidate effects resulting in the observed downregulation of AKT as IRS1 is a target protein of the negative feedback regulation via S6K (Tremblay et al, 2007; Zhang et al, 2008).

The development of mechanistic models provides a tool to understand complex biological questions and to gain insights into molecular mechanisms regulating dynamic behavior. Accordingly, based on mathematical modeling of the HGF signal transduction pathway cross-talk, we dissected dependencies of MAP kinase and PI3 kinase signal transduction in primary hepatocytes and established the link to the regulation of cell cycle progression (D'Alessandro et al, 2015; Mueller et al, 2015). Further, Dalla Pezze

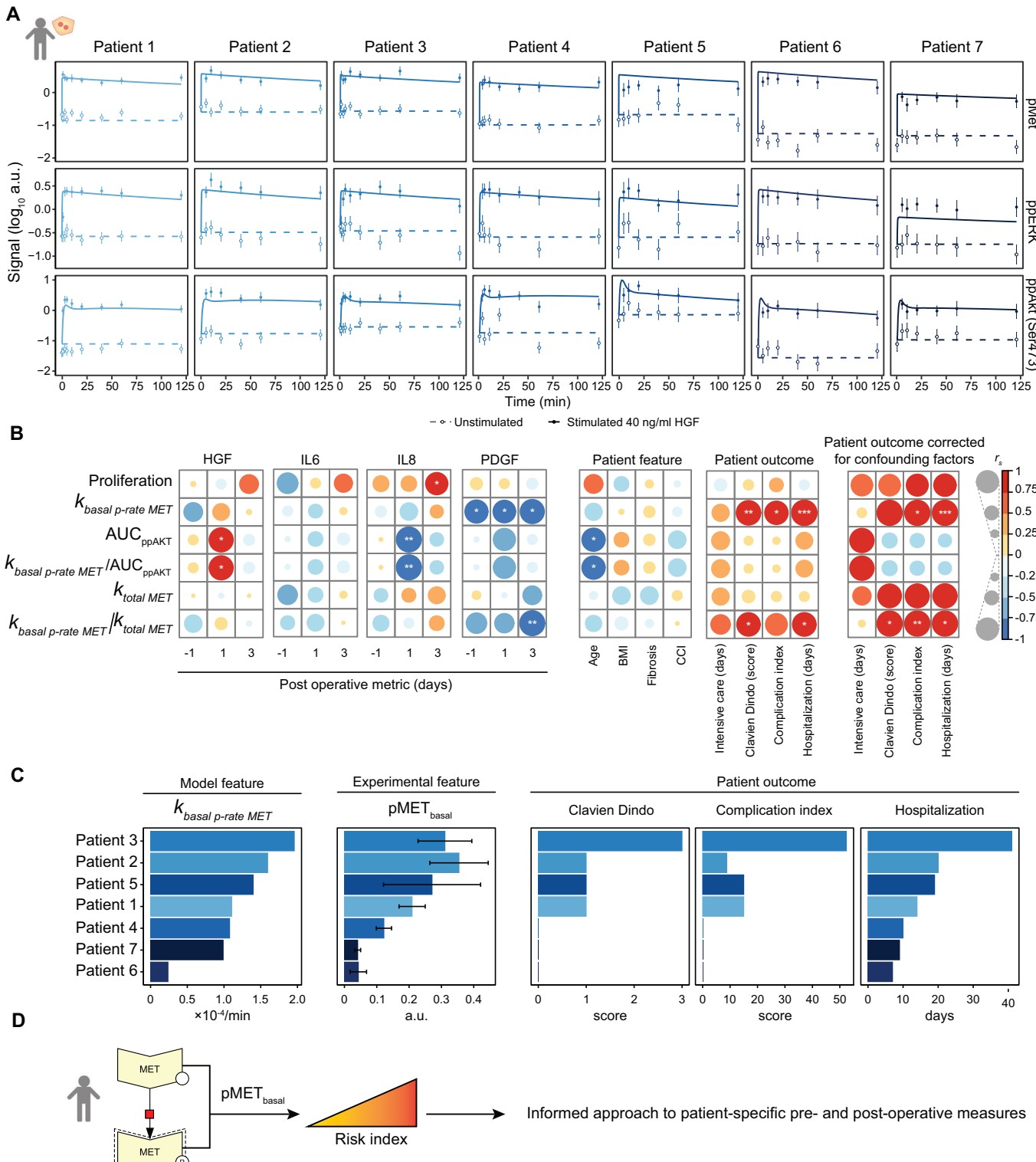

et al resolved the interactions of the mTOR pathway upon insulin stimulation (Dalle Pezze et al, 2012). Based on these findings, we included in our model mTOR as a metabolic gateway in the liver (Jia et al, 2014) to capture the Western diet-induced alterations in HGF-dependent signal transduction in primary hepatocytes. Interestingly, the only reaction rate, which we identified with our mathematical modeling approach as dysregulated in this disease scenario, was the basal phosphorylation of the MET receptor. Further analysis of the mathematical model showed that the reaction rate of basal MET phosphorylation was sufficient to describe the observed diet-induced changes in AKT phosphorylation and MAPK signal transduction. These results disentangle the

**Figure 6. Basal pMET levels in primary human hepatocytes correlate to patient outcome.**

(A) Time-resolved immunoblot measurements and model fits for pMET, pERK and ppAKT in primary human hepatocytes derived from seven patients. Cells were stimulated with 40 ng/ml HGF or left untreated. Signal is shown in $\log_{10}$ arbitrary units (a.u.). Data points are displayed as dots along with error bars representing $1\sigma$ confidence interval estimated from technical replicates ($n = 1$–3 per patient) using a combined scaling and error model. Model trajectories are depicted as lines. (B) Spearman correlation of hepatocyte proliferation and model features with post-operative blood metrics, patient-specific features and outcome. Significance levels were calculated using the algorithm AS89 (Best and Roberts, 1975) and are indicated as *$p < 0.05$, **$p < 0.01$, ***$p < 0.001$. BMI body mass index, CCI Charlson comorbidity index. In addition, partial correlation coefficients were calculated for the patient outcome correcting for the confounding factors age, BMI, fibrosis and CCI. (C) The patient-specific basal phosphorylation rate of MET as estimated by the model is depicted in comparison to the experimentally measured basal pMET levels, obtained as mean of all unstimulated pMET measurements per patient in arbitrary units (a.u.). In comparison, the clinical metrics Clavien Dindo, complication index and hospitalization are shown. Patients are sorted by the basal phosphorylation rate of MET, colors indicate patient number. Error bars represent the standard deviation of 7–9 replicates. (D) The basal phosphorylation of MET as an informative metric in patients for liver disease burden and patient recovery after liver surgery. The proposed use as risk index gives an informed approach to patient-specific pre- and post-operative measures. Source data are available online for this figure.

complex interrelations of the impact of metabolic alterations and proliferative signal transduction in hepatocytes and thus shed new light on molecular dysregulation in the context of chronic liver diseases. So far, the analysis of HGF-induced signal transduction in the liver has primarily focused on its role in repair of liver damage and regeneration (Oe et al, 2005), whereas the impact of metabolic alterations, which could be critical at early stages of chronic liver diseases, has not yet been considered. Importantly, we uncover that the exposure to the Western diet increases not only the levels of basal MET phosphorylation but also results in an increased ex-vivo proliferation of primary hepatocytes. These insights suggest that although the early steatotic phenotype in liver disease may be reversible, the metabolic state of the hepatocytes and their capacity to interpret external signals has shifted. We have not directly addressed the molecular mechanism driving the increased basal phosphorylation of MET, but hypothesize that the diet-induced changes in the cell surface membrane of hepatocytes may alter the dimerization rate of MET receptors and change the heterodimerization capabilities of MET rendering the receptor a central integrator of metabolic changes. In support of this hypothesis, it has been reported that a high fat diet decreased the cholesterol content of the plasma membrane of hepatocytes and reduced the affinity of the insulin receptor to insulin (Sabapathy et al, 2022). In line with this report, our analysis of canonical pathways showed that cholesterol biosynthesis was downregulated in WD-derived hepatocytes. Since HGF has been identified as modulator of liver fibrosis, which develops as sustained liver disease (Kwiecinski et al, 2011; Tekkesin et al, 2011), we propose that HGF-induced signal transduction determines the fate of hepatocytes and based on our results the extent of basal MET phosphorylation could be a central indicator facilitating the quantification of the alterations.

The liver is organized in hexagonal liver lobules consisting of plates of hepatocytes interspersed by small capillaries, the liver sinusoids. Blood enters the hexagonal lobules through portal veins and drains towards a central vein, establishing a gradient of nutrients, metabolites and oxygen that correlates with distinct enzymatic equipment of hepatocytes and results in metabolic zonation of the liver lobule. As steatosis primarily affects the pericentral zone it is conceivable that susceptibility to metabolic insults might differ. We attempted to address the distribution of basal phosphorylated MET by histological stainings, but so far, the results remained inconclusive and require further optimization.

Previous reports indicated that liver steatosis affects the regenerative capacity of the liver (Allaire and Gilgenkrantz, 2018; Ghanemi et al, 2020). Therefore, our observations could be of relevance for patients undergoing liver surgery. To assess this, we analyzed primary hepatocytes isolated from seven patients undergoing liver hepatectomy to adapt our dynamic pathway model of HGF signal transduction to the human situation. In line with our previous studies (Dehlke et al, 2022; Murtha-Lemekhova et al, 2021), we observed little correlation ($p > 0.01$) of HGF, IL6, IL8, PDGF, age, BMI, and fibrosis with the patient outcome, which was assessed by Clavien Dindo score, complication index and hospitalization days. In contrast, we uncovered that the basal phosphorylation rate of MET strongly correlated with the patient outcome. The importance of the basal MET phosphorylation rate in this context was further supported by the immunoblot based quantifications of the pMET level in the patient-derived hepatocytes that followed the trend of the basal MET phosphorylation rate and the clinical outcome. Furthermore, it has been reported that hepatocytes undergo metabolic reprogramming during liver proliferation (Chembazhi et al, 2021). It was proposed that since proliferating hepatocytes cannot sustain liver-specific metabolic functions, other hepatocytes shift into a hyperactive metabolic state. These notions could explain why patients, with higher basal MET phosphorylation, undergoing hepatectomy are less capable to compensate the metabolic function in the liver and show higher risk of hepatic failure. Nevertheless, we acknowledge that a cohort of seven patients is very limited and our observations should be interpreted with caution.

In conclusion, our model-based insights identify the basal phosphorylation rate of MET as an indicator of alterations in the metabolic state of hepatocytes and its impact on proliferative signal transduction. Interestingly, we uncover a strong correlation of the model parameter of the basal MET phosphorylation rate and the clinical outcome upon liver surgery. Since the extent of basal MET phosphorylation is a similarly good predictor of the clinical outcome as the estimated rate, the quantification of MET phosphorylation in surgical liver samples might provide a readily accessible readout to predict the clinical outcome. Taken together, patient-specific pMET levels could be exploited to assess the health status of the liver and to estimate the risk of a patient to suffer from liver failure after surgery.

## Methods

### Mouse housing and feeding

Standard diet control mice: Male C57BL/6N (Charles River, RRID MGI:2159965) control mice were housed at the German Cancer

Research Center (DKFZ) animal facility under a constant light/dark cycle and allowed ad libitum access to water and food and maintained on a standard mouse diet (KLIBA NAFAG 3437). The experiments were approved by the governmental review committee on animal care of the state Baden-Württemberg, Germany (reference number G-14/17 and G-33/17).

Western diet-fed mice: Male C57BL/6N mice (Charles River, RRID MGI:2159965) of 8 weeks of age were housed at the Leibniz-Institut für Arbeitsforschung an der Technischen Universität Dortmund (IfaDo) under a constant light/dark cycle and allowed ad libitum access to water and food and maintained on a Western diet (Research Diet Inc., D09100301, including trans fats) containing 40% kcal fat, 40% kcal carbohydrates and 2% weight cholesterol for 12 or 13 weeks. The experiments were approved by the governmental review committee of animal care of the state Nordrhein-Westfalen, Germany (reference number 84-02.04.2017.A177).

Transgenic R26p-Fucci2 mice: R26p-Fucci2 mice were obtained from RIKEN Center for Developmental Biology (CDB, RRID MGI:5491593) and recovered by embryo transfer. The mice were housed at the German Cancer Research Center (DKFZ) animal facility under a constant light/dark cycle and allowed ad libitum access to water and food. Heterozygous R26p-Fucci2 mice were bred with wild type C57BL/6N mice to maintain the line. Only male animals were used for the performed analyses. Animals were either maintained on a standard mouse diet (KLIBA NAFAG 3437) or switched to a Western diet (WD, Research Diet Inc., D16022301, no trans fats) at the age of 8 weeks and fed with WD for 12 weeks. The experiments were approved by the governmental review committee of animal care of the state Baden-Württemberg, Germany (reference number A24/10, G-14/17, G-33/17).

## Isolation of primary mouse hepatocytes

Mice of final age of 20–21 weeks were used for primary mouse hepatocyte isolation. Hepatocytes were isolated according to a standardized procedure with adaptations to improve yields for WD-fed mice (Mueller et al, 2015). Anesthesia was carried out by intraperitoneal injection of 11.25 mg per 100 mg body weight ketamine hydrochloride (100 mg/ml, zoetis), 1.6 mg per 100 mg body weight xylazine hydrochloride (2% (w/v), Bayer HealthCare) and 1.5 mg per 100 mg acepromazine (cp-pharma). The abdominal cavity was opened and the vena cava inferior or portal vein was cannulated with a 24 G venous catheter to enable perfusion of the liver. The liver was perfused with EGTA-containing buffer (0.6% (w/v) glucose, 105 mM NaCl, 2.4 mM KCl, 1.2 mM $KH_2PO_4$, 26 mM Hepes, 490 M L-glutamine (Gibco), 512 µM EGTA, 15% (v/v) amino acid solution (270 mg/l L-alanine, 140 mg/l L-aspartic acid, 400 mg/l L-asparagine, 270 mg/l L-citrulline, 140 g/l L-cysteine hydrochloride monohydrate, 1 g/l L-histidine monohydrochloride monohydrate, 1 g/l L-glutamic acid, 1 g/l L-glycine, 400 mg/l L-isoleucine, 800 mg/l L-leucine, 1.3 g/l L-lysine monohydrochloride, 550 mg/l L-methionine, 650 mg/l L-ornithine monohydrochloride, 550 mg/l L-phenylalanine, 550 mg/l L-proline, 650 mg/l L-serine, 1.35 g/l L-threonine, 650 mg/l L- tryptophane, 550 mg/l L-tyrosine, and 800 mg/l L-valine; pH 7.6) ; pH 8.3) for 5 min and collagenase-containing buffer (0.6% (w/v) glucose, 105 mM NaCl, 2.3 mM KCl, 1.2 mM $KH_2PO_4$, 25 mM Hepes, 490 µM L-glutamine (Gibco), 5.3 mM $CaCl_2$, 12% (v/v) amino acid

solution, 444 µg/ml collagenase type 1-A; pH 8.3) for up to 10 min at a flow rate of 8 ml/min. The portal vein or vena cava inferior was incised to allow sufficient buffer outflow. Following perfusion, the liver was withdrawn and transferred into suspension buffer (0.6% (w/v) glucose, 105 mM NaCl, 2.4 mM KCl, 1.2 mM $KH_2PO_4$, 26 mM Hepes, 1 mM $CaCl_2$, 0.4 mM $MgSO_4$, 0.2% (w/v) BSA, 490 µM L-glutamine (Gibco), 15% (v/v) amino acid solution; pH 7.6). Hepatocytes were isolated by disrupting the liver capsule and filtering the resulting cell suspension through a 100 µm cell strainer. Cells were washed by centrifugation at $50 \times g$ for 5 min at 4 °C or twice at $50 \times g$ for 2 min at room temperature and resuspended in adhesion medium (phenol red-free Williams E medium (PAN Biotech) supplemented with 10% (v/v) FCS (Gibco), 0.1 µM dexamethasone, 0.1% (v/v) insulin, 2 mM L-glutamine, 1% (v/v) penicillin/streptomycin (Gibco)). Cell yield and vitality were determined by Trypan Blue staining using a Neubauer counting chamber. Preparations with vitality greater than 70% were employed for experiments. For each experiment, cells from a single mouse were used, or if necessary, cells were pooled from up to three mice.

## Collection of patient tissue samples

Primary human hepatocytes were isolated from samples taken from the specimen of patients undergoing major hepatectomy at the Department of General, Visceral, and Transplantation Surgery of Heidelberg University Hospital. All patients were screened for eligibility irrespective of the indication for major hepatectomy and included provided they have signed the valid informed consent form. Informed consent of the patients for the use of tissue for research purposes was obtained corresponding to the ethical guidelines of University Hospital Heidelberg (reference number S-557/2017). A non-tumor tissue sample with intact Glisson's capsule, weighing approximately 20 g was collected and immediately transported in William's E medium (PAN Biotech) to the laboratory for further processing.

## Isolation of primary human hepatocytes

Cell isolation took place under a fume hood in sterile conditions using a standardized protocol adapted from Kegel et al (2016). Three-to-8 cannulas were placed in the vessels of the non-capsuled surface of the liver sample. The cannulas were fixed with Histoacryl tissue glow and the blood was flushed from the tissue. The liver tissue was perfused with 500 ml 39 °C 1× Perfusion Solution (142 mM NaCl, 6.7 mM KCl, 10 mM HEPES 12.5 mM EGTA, 6.25 mM N-acetyl-L-cystein) using a flow rate of 60–70% for 20–30 min until the liver tissue became light yellow. Afterwards the perfusion fluid was changed to 39 °C digestion solution (33.5 mM NaCl, 3.35 mM KCl, 50 mM HEPES, 0.25% BSA, 10% FCS, 1 mg/ml Collagenase P) for 15 min. To stop the digestion the liver sample was rinsed with ice-cold Stop Solution (20% FCS in DPBS (PAN Biotech)). After removing the cannulas, a scalpel was used to open the liver tissue. By flushing with Stop Solution and shaking the tissue gently, the cells were released from the tissue. The cell suspension was collected and filtered through a 100 µm cell strainer to 50 ml plastic falcons. The cell suspension was centrifuged at $50 \times g$, 5 min, 4 °C. The cell pellet was washed with DPBS (PAN Biotech) followed by another round of centrifugation. Afterwards,

the pellet was resuspended in adhesion medium. Cell yield and vitality were determined by Trypan Blue staining using a Neubauer counting chamber.

## Collection of human blood samples and plasma extraction

The blood of the patients was taken at the University Hospital Heidelberg one day before and 1, 3 and 7 days after surgery. Informed consent of the patients for the use of blood for research purposes was obtained corresponding to the ethical guidelines of University Hospital Heidelberg (reference number S-557/2017). For collecting blood, a vein in the arm bend was punctured, after a two-time disinfection with an alcoholic skin antiseptic. The blood was collected in EDTA 2.7 ml (Sarstedt). To receive the blood plasma the blood samples were centrifuged at full speed briefly and the supernatant was collected and frozen in $-80\,°C$ until further use.

## Analysis of patient plasma cytokines

To analyze the preoperatively and postoperatively collected blood plasma samples of the patients, Bio-Plex Pro Cytokine, Chemokine, and Growth Factor Assay (Biorad) and the Bio-Plex Pro TGFβ Assay (Biorad) was used for multiplexing according to the standardized protocol established by Biorad. For multiplexing, HGF, IL6, IL8, PDGF, TGFβ1 and TNFα were analyzed. The blood plasma was thawed on ice and centrifuged twice ($1000 \times g$, 15 min at $4\,°C$). The samples were diluted 1:4 using Bio-Plex sample diluent. 50 µl coupled beads mix were pipetted to each well of the assay plate after vortexing for 30 s at medium speed. Then the beads were washed twice with the Bio-Plex wash buffer. After vortexing, 50 µl of the pre-diluted standards and samples were added. Samples and standards were assayed in technical duplicates. The plate was covered with sealing tape to protect from light and incubated on a shaker for 30 min at $850 \pm 50$ rpm. After the incubation, the beads were washed three times with washing buffer before 25 µl of the detection antibody mix was added. The plate was covered with sealing tape and incubated for a second time on the shaker for 30 min at $850 \pm 50$ rpm at room temperature. The incubation was followed by three more washing steps and the addition of 50 µl SA-PE, which was incubated for 10 min at $850 \pm 50$ rpm for 10 min. After the final three washing steps the beads were re-suspended in 125 µl assay buffer and were shaken at $850 \pm 50$ rpm for 30 s at room temperature and then measured with a Bio-Plex 200 reader.

For the TGFβ kit, after a 15 min centrifugation at $1000 \times g$ and 10 min at $10,000 \times g$ both at $4\,°C$, acid (1 M HCl) and samples were added in 1:5 proportions and incubated at room temperature for 10 min. The samples were neutralized by adding 1 volume of base (1.2 M NaOH/0.5 M HEPES) and vortexing. The (untreated) sample was in total diluted 1:16 with Bio-Plex sample diluent. After adding 50 µl of magnetic beads, washing two times, and pipetting 50 µl of standards and samples the plate was covered with sealing tape and incubated on a shaker at $850 \pm 50$ rpm for 2 h at room temperature. The incubation was followed by three more washing steps and the addition of 25 µl antibodies, which were incubated for 1 h at $850 \pm 50$ rpm at room temperature. The samples were washed three times and incubated with 50 µl of SA-PE for 30 min at $850 \pm 50$ rpm at room temperature. After washing three times again the beads were re-suspended in 125 µl assay

buffer at each well and incubated for 30 s at $850 \pm 50$ rpm before the plate was measured with a Bio-Plex 200 reader.

## Collection of patient characteristics

Patient data (Dataset EV5) were prospectively collected by extraction from the electronic patient record system of the Heidelberg University Hospital. The collected information included clinic demographic data, preoperative clinical course, laboratory values, intraoperative variables, complications, time to event data, diagnose-related information, pre-operative treatment and histopathology reports. From this data, the following scores were obtained: Clavien Dindo score (Dindo et al, 2004), Charlson comorbidity index (Charlson et al, 1987) and comprehensive complication index (Slankamenac et al, 2013).

## Cultivation and stimulation of primary hepatocytes

Isolated primary hepatocytes were plated on collagen I-coated cell ware in adhesion medium. Using primary mouse hepatocytes, for the dose response and time course experiments $2 \times 10^6$ cells were seeded per 6 cm dish. For live cell imaging, $7.5 \times 10^3$ primary mouse hepatocytes were seeded per well of a 96-well plate. For time course experiments with primary human hepatocytes, $2.5 \times 10^6$ cells were seeded per 6 cm dish. For proliferation experiments, 150,000 cells per well were seeded on 6-well plates. Following plating, cells were allowed to adhere for 4 h (SD mouse hepatocytes) or 4–6.5 h (WD mouse hepatocytes and human hepatocytes). Subsequently, hepatocytes were washed, twice vigorously and once gently, with PBS (PAN Biotech) to remove unattached cells and cultured in serum-free medium overnight. During all incubation times, the cells were cultured at $37\,°C$, 5% $CO_2$ and 95% relative humidity.

For dose response and time course experiments, cells were washed gently three times and supplied with serum- and dexamethasone-free medium for 6 h. Primary mouse hepatocytes were stimulated with 0.1, 1, 2, 4, 10, 20, 40, 80, 100 and 120 ng/ml of recombinant mouse HGF (rmHGF, R&D systems) for 10 min for dose response experiments and with 40 ng/ml HGF for 5, 10, 20, 40, 60, 120, 180, and 240 min for time course experiments. Unstimulated control plates were taken out of the incubator together with stimulated plates and a zero dose / time point was taken in addition.

Primary human hepatocytes were stimulated with 40 ng/ml recombinant human HGF (R&D systems) for 1, 3, 5, 10, 20, 40, 60, and 120 min. Unstimulated control plates were taken out of the incubator together with stimulated plates and a zero time point was taken in addition.

## Live cell imaging of the cell cycle progression in primary mouse hepatocytes

Seeded primary mouse hepatocytes from R26p-Fucci2 transgenic mice were supplemented with purified AAV encoding Histone2B-mCerulean produced using a triple transfection protocol by Dirk Grimm (Heidelberg University) (Grimm, 2002) for the duration of cell adhesion. Following adhesion, cells were washed twice with serum-free medium and incubated in serum-free medium for 24 h before stimulation. Shortly before time-lapse microscopy, media were changed to 100 µl fresh serum-free medium supplemented

with 40 ng/ml HGF. Hepatocytes were imaged with a Nikon Eclipse Ti Fluorescence microscope controlled with NIS-Elements software. Temperature (37 °C), $CO_2$ (5%) and humidity were held constant by an incubation chamber enclosing the microscope and a 96-well plate stage insert. Four channels were acquired: brightfield channel, CFP channel (Histone2B-mCerulean), RFP channel (mCherry-hCdt1), and YFP channel (mVenus-hGeminin). Time lapse microscopy was performed for up to 65 h with a sampling rate of 20 min.

Image analysis was performed using the Fiji software. Background subtraction was performed with rolling ball method. CFP channel was used for segmentation of nuclei using a standard threshold-based algorithm. Mean RFP and YFP intensity was quantified for all segmented nuclei at each time frame.

## Snapshot population analysis of Fucci2 data

The programming language R was used to perform subpopulation analysis of Fucci2 data. All segmented nuclei within a specified time window between 42 and 54 h were displayed as a scatter plot to quantify percentage of subpopulations in a given cell cycle phase. Four subpopulations were defined using arbitrary thresholds for mCherry and mVenus signals based on unstimulated condition as a reference. The four compartments were defined as RFPhigh/YFPlow (G1-phase, red), RFPhigh/YFPhigh (S-phase, orange), RFPlow/YFPhigh (G2-phase, green), RFPlow/YFPlow (M-phase, gray).

## Time-resolved analysis of cell cycle progression in single cells

Manual segmentation and tracking were performed to obtain single cell tracks. To perform reliable quantification of cell cycle duration and cell cycle entries, only cells that were present in the field of view for the whole duration of the time lapse were considered. A region of interest (ROI) was defined in individual nuclei in the image of the CFP channel (Histone2B-mCerulean) for each frame. At the event of cell division, only one of the daughter cells was followed until the end of the whole time-lapse experiment. The obtained set of ROIs was applied to the RFP (mCherry-hCdt1) and YFP (mVenus-hGeminin) channels to extract the RFP and YFP mean intensity values after background subtraction by rolling ball method. The two thresholds for the RFP and YFP channels defined in the snapshot analysis were applied to assign cells to the four cell cycle phases, represented as the respective color in the heatmap. To account for fluctuations in the expression of the Fucci2 fluorescent probes, cell cycle entries were considered as transition from S to G2-phase.

## Cell lysis

Cells were lysed in 500 μl (mouse hepatocytes) or 300 μl (human hepatocytes) RIPA buffer (50 mM Tris pH 7.4, 150 mM NaCl, 1 mM EDTA, 1 mg/ml Deoxycholic acid, 0.5 mM $Na_3VO_4$, 2.5 mM NaF, 1% NP40, 0.1% AEBSF, 0.1% AP) on ice, incubated while rotating at 4 °C for 20 min and were sonicated for 30 s (Amplitude: 80%, 0.1 on, 0.5 off) (human hepatocytes only) followed by centrifugation at 4 °C for 10 min at 20 817 × g. The supernatant representing protein lysates were stored at −80 °C. Pierce BCA

Protein Assay Kit (Thermo Fisher) was used to determine the protein concentration in the total cell lysates.

## Quantitative immunoblotting

| Target | Species | Dilution used | Manufacturer | Catalog number |
|---|---|---|---|---|
| For primary mouse hepatocyte experiments | | | | |
| pMET Tyr1234/ 1235 | Rabbit | 1:2000 | Cell Signaling | #3077 |
| MET | Mouse | 1:500 | Santa Cruz | #8057 |
| pAKT Thr308 | Rabbit | 1:2000 | Cell Signaling | #4056 |
| pAKT Ser473 | Rabbit | 1:2000 | Cell Signaling | #4058 |
| AKT | Rabbit | 1:2000 | Cell Signaling | #9272 |
| pERK Thr202/ Tyr204 | Rabbit | 1:10,000 | Cell Signaling | #9101 |
| ERK | Rabbit | 1:5000 | Cell Signaling | #9102 |
| pS6 Ser235/236 | Rabbit | 1:5000 | Cell Signaling | #2211 |
| S6 | Mouse | 1:2000 | Cell Signaling | #2317 |
| For primary human hepatocyte experiments | | | | |
| pMET | Rabbit | 1:2000 | Cell Signaling | #3129 |
| MET | Mouse | 1:2000 | Cell Signaling | #3148 |
| pAKT | Rabbit | 1:1000 | Cell Signaling | #4058 |
| AKT | Rabbit | 1:2000 | Cell Signaling | #9272 |
| pERK | Rabbit | 1:1000 | Cell Signaling | #9101 |
| ERK | Rabbit | 1:1000 | Cell Signaling | #9102 |
| pS6K Thr 389 | Rabbit | 1:2000 | Cell Signaling | #9234 |
| pS6K | Rabbit | 1:2000 | Cell Signaling | #2708 |
| Secondary antibodies | | | | |
| HRP anti Rabbit | Goat | 1:10,000 | Jackson ImmunoResearch | 111-035-144 |
| HRP anti Mouse | Goat | 1:10,000 | Jackson ImmunoResearch | 115-035-146 |

For total cell lysate analysis 30 μg protein (mouse hepatocytes) or 20 to 30 μg (human hepatocytes) was filled up to 25 μl total volume with RIPA buffer, mixed with 25 μl 2× sample buffer (4% SDS, 100 mM tris-HCl pH 7.4, 20% glycerol, 200 mM DTT, bromophenol blue, 10% β-mercaptoethanol), incubated for 3–5 min at 95 °C and used for quantitative immunoblotting. Total cell lysates and immunoprecipitation samples were loaded on 10% polyacrylamide gels in a randomized order to avoid correlated blotting errors (Schilling et al, 2005). Magic Marker (Invitrogen) and Precision Plus Protein Standard (Biorad) were loaded in addition. The proteins were separated by discontinuous gel electrophoresis (SDS-PAGE) (40 mA per gel for 3 h) covered with Laemmli Buffer (192 mM glycin, 25 mM Tris, 0.1% SDS). The separated proteins were transferred in a semi-dry system onto PVDF membranes (Immobilon P, Merck Millipore) using transfer buffer (192 mM glycine, 25 mM Tris, 0.075% SDS, 0.5 mM Na3VO4, 15% EtOH) (Hoefer TE77 semi-dry transfer system (250 mA per blot for 1 h)). The membrane was blocked by drying after being soaked in ethanol followed by a 10 min reactivation in

TBS-T (10 mM Tris pH 7.4, 150 mM NaCl, 0.2% Tween-20). The membranes were incubated with primary antibodies (against MET (Sanza Cruz and Cell Signaling), pMET Tyr 1234/1235, pAKT Thr 308, pAKT Ser 473, AKT, pERK Thr 202/Tyr 204, ERK, pS6K Thr 389, S6K, pS6 Ser 235/236, S6 (all Cell Signaling)) in 2.5% BSA in TBS-T overnight at 4 °C.

The membranes were washed twice with TBS-T for 5 min followed by a 1 h incubation with secondary Antibody (HRP Goat Anti-Rabbit (Dharmacon) or HRP Goat Anti-Mouse (Dharmacon), 1:10 000 in 2.5% BSA in TBS-T). After two additional washing steps with TBS-T and one with TBS the blots were developed by incubation with Amersham ECL Western Blotting Detection Reagents (GE Healthcare) for 2 min and detected on an ImageQuant (GE Healthcare). The proteins were quantified using ImageQuant TL software (GE Healthcare, version 7.0). For further detection, the blots were treated with stripping buffer (62.5 mM Tris pH 6.8, 2% SDS, 0.7% β-mercaptoethanol) for 20 min at 65 °C to remove the bound antibodies. After washing with double-distilled water, blocking by drying, and reactivation by shaking in TBS-T, membranes were employed for further antibody binding. Alternatively, the secondary antibody was inactivated using 30% $H_2O_2$ for 15 min at 37 °C followed by three washing steps in double-distilled water for 5 min each and reactivation in TBS-T.

## Proliferation assay

At time point 0 h, primary human hepatocytes were washed with PBS twice. For proliferation, primary human hepatocytes were stimulated by adding new serum and dexamethasone-free medium containing 40 ng/ml recombinant human HGF. Medium without HGF was used for control cells. After 48 h the hepatocytes were washed once with PBS and the plates were frozen at −20 °C. Control cells were washed with DPBS twice at time point 0 h and frozen at −20 °C. For DNA content measurement, staining with SybrGreen I (Invitrogen, S7563) was performed. The frozen cells were treated with 2 ml SybrGreen I working solution (PBS, 1:100 Triton X 100, 1:2500 SybrGreen I) per well, incubated protected from light for 1 h. Signal emission was measured at 485 nm excitation using a Tecan Infinite Pro 200 reader. The optimal gain was determined with the plate for which the highest signal was expected and this value was used for all further measurements.

## Proteomics sample preparation

Lysates of unstimulated primary mouse hepatocytes were used for proteome analysis. Protein concentrations were determined by employing the BCA Protein Assay Kit (Pierce,Thermo Fisher). In total, 20 μg of protein per sample was used for further processing. The disulfide bonds of proteins were reduced with 40 mM Tris(2-carboxyethyl) phosphine (TCEP) and then alkylated with 10 mM chloroacetamide (CAA) for 60 min at 37 °C. Protein digestion and clean-up were performed using an adapted version of the automated paramagnetic bead-based single-pot, solid-phase-enhanced sample-preparation (Auto-SP3) protocol (Muller et al, 2020) on the Bravo liquid handling platform (Agilent). Briefly, for bead preparation, Sera-Mag Speed Beads A and B (Ge Healthcare) were vortexed until the pellet was dissolved. The suspension was placed on a magnetic rack, and after one minute, the supernatant was removed. The beads were taken off the magnetic rack and

suspended in water. This procedure was repeated three times. A total of 10 μl of bead A was combined with 10 μl of bead B, and the final volume was corrected to 100 μl with $H_2O$. To each sample, a total of 5 μl of A + B beads were added. To induce the binding of the proteins to the beads, ethanol was added to each sample to a final 50% concentration (v/v). Samples were then incubated for 15 min at room temperature and 800 rpm. After the incubation step, samples were placed again on a magnetic rack, and after one minute, the supernatant was removed. Samples were taken off the magnetic rack and suspended in 80% ethanol. This procedure was repeated three times. Finally, samples were reconstituted in 100 mM TEAB buffer containing trypsin (enzyme/protein ratio of 1:25) and digested overnight on a shaker at 37 °C and 1000 rpm. After digestion, the recovered peptides were dried by vacuum centrifugation and stored at -80 °C until use.

## LC-MS/MS analysis

Nano-flow LC-MS/MS was performed by coupling an Ultimate 3000 HPLC (Thermo Fisher Scientific, USA) to a Orbitrap Exploris mass spectrometer (Thermo Fisher Scientific, Germany). Peptide samples were dissolved in 15 μl loading buffer (0.1% formic acid (FA), 2% ACN in MS-compatible $H_2O$), and 2 μl were injected for each analysis. The samples were loaded onto a pre-column (trap) at higher flow rates (PEPMAP 100 C18 5 μm 0.3 mm × 5 mm, Thermo Scientific), using a loading pump and then a valve was switched to delivered to an analytical column (75 μm × 30 cm, packed in-house with Reprosil-Pur 120 C18-AQ, 1.9 μm resin, Dr. Maisch) at a flow rate of 3 μl/min in 98% buffer A (0.1% FA in MS-compatible $H_2O$). After loading, peptides were separated using a 120 min gradient from 2% to 38% of buffer B (0.1% FA, 80% ACN in MS-compatible $H_2O$) at 350 nl/min flow rate. The Orbitrap Exploris 480 was operated in data-independent (DIA) mode, with a m/z range of 350–1400. Full scan spectra were acquired in the Orbitrap at 120,000 resolution after accumulation to the set target value of 300% (100% = 1e6) and maximum injection time of 45 ms. The full scans were followed by DIA scans. A total of 47 isolation windows were defined, with a m/z range of 406–986. DIA scan spectra were acquired at 30,000 resolution after accumulation to the set target value of 1000% (100% = 1e5) and maximum injection time of 54 ms. Normalized collision energy (NCE) was set to 28%.

## Database search and data analysis

All DIA–MS data files were analyzed with a directDIA workflow using Spectronaut 15.5 (Biognosys, Zurich, Switzerland). For the Pulsar search, the UniProt human-reviewed canonical reference proteome (20,593 entries, UP000005640 downloaded on December 7th, 2022) and the UniProt mouse-reviewed canonical reference proteome (20,404 entries, downloaded on December 7, 2022) were used. The default settings for database match include: full specificity trypsin digestion, peptide length of between 7 and 52 amino acids and maximum missed cleavage of 2. N-terminal methionine was removed during preprocessing of the protein database. Carbamidomethylation at cysteine was used as a fixed modification, protein N-terminal acetylation and methionine oxidation were set as variable modifications. The false discovery rates (FDRs) were set as 0.01 for the peptide-spectrum match (PSM), peptide and protein identification. Other Spectronaut parameters for identification included maximum intensity

as MZ extraction strategy, 0.01 Precursor Qvalue Cutoff, 0.2 Precursor PEP Cutoff, stripped sequence as single hit definition. For quantification, identified (Qvalue) was set for precursor filtering, MaxLFQ as protein LFQ method and MS2 quantification with area as quantity type. For further analysis, only those proteins without missing values were considered. For the analysis of the proteome data the public server at usegalaxy.org was used (Afgan et al, 2016). Analysis was based on the R package *limma* (Ritchie et al, 2015). Statistical testing of significance was performed using the *voom* function from Bioconductor (Law et al, 2014). Significance was considered with adjusted $p$ value < 0.05. Functional annotation of canonical pathways was performed with the use of Ingenuity Pathway Analysis (Krämer et al, 2014) by QIAGEN. Thresholds were set to an adjusted $p$ value of 0.05 and a $\log_2$ fold change $<-0.5$ or $>0.5$.

## Data processing and estimation of uncertainties

Immunoblot data from individual experiments were scaled and thereby aligned for each target using the R package *blotIt* (Kemmer et al, 2022). Uncertainties, corresponding to $1\sigma$ confidence intervals, were estimated along with scaling factors for each experiment using the scaling model $Y_{ij} \sim \frac{y_i}{s_j}$ with measurements $Y_{ij}$, scaling factors $s_j$ and scaled values $y_i$, combined with a relative error model $\sigma_{ij} \sim e_{rel} \cdot \frac{y_i}{s_j}$ with error parameter $e_{rel}$ and estimated errors $\sigma_{ij}$. The biological effects $i$ were distinguished for individual targets, measurements time points, mouse diets and HGF stimulation doses. Scaling effects $j$ were distinguished per experiment and target. For the mouse model 4466 raw data points were aligned by *blotIt* to obtain 772 data points with confidence intervals that served for calibration of the mouse model. In addition, 209 raw data points for protein abundances obtained from DIA measurements were $\log_2$ transformed and averaged to obtain 22 data points with calculated standard deviations. Finally, the number of molecules per cell of AKT determined by quantitative immunoblotting was also utilized for model calibration. For patient-derived primary human hepatocytes, 2443 raw data points were aligned by *blotIt* to obtain 1106 data points with confidence intervals that served for calibration of the human model. In addition, 112 raw data points for protein abundances obtained from DIA measurements were $\log_2$ transformed and averaged to obtain 56 data points with calculated standard deviations.

## Parameter estimation and model development for primary mouse hepatocytes

The mathematical modeling was performed in the R package *dMod*, which provides an environment for the development of ODE models, parameter estimation and uncertainty analysis (Kaschek et al, 2019). The final ODE model was composed of 23 species and 26 reactions derived from the law of mass-action (Dataset EV1). Measurements were mapped to model states by means of observables as displayed in Dataset EV2. The estimated parameter set corresponding to the global optimum along with profile-likelihood based confidence intervals is listed in Dataset EV3. Log-transformation of parameters was used to ensure positivity and numerical stability. A set of parameter transformations (Dataset EV4) was used to incorporate the calculation of analytical steady-state expressions (Rosenblatt et al, 2016) and reformulations necessary for model reduction (Maiwald et al, 2016). Parameter

values were estimated using the maximum likelihood method, performing a deterministic multi-start optimization with the trust region optimizer (R package *trust*) and starting from 250 randomly chosen parameter sets. Results of the optimization run for the final model are displayed as waterfall plot (Raue et al, 2013) in Fig. EV2D. The global optimum was found in 164 out of the 250 starts for the final model. The profile likelihood (Raue et al, 2009) was used to assess the identifiability of parameters and determine confidence intervals for the estimated values of 22 initial protein concentrations, 13 scaling and offset parameters, and 27 dynamical parameters of the mouse model. For the identification of mechanistic differences in the signal transduction of SD and WD hepatocytes models differing in their number and the selection of diet-specific parameters were compared using the Bayesian information criterion (Schwarz, 1978).

## Application of the mouse model to primary human hepatocytes

The calibrated mouse model was used for the analysis of measurements derived from primary human hepatocyte. The parameter values were fixed to the estimates obtained for the mouse data except for a parameter subset that was identified by model comparison based on the Bayesian information criterion. These human-specific parameters comprise the basal MET phosphorylation rate and protein abundances, which were estimated as patient-specific, scaling and offset parameters, as well as the HGF-induced MET phosphorylation rate and the pMET degradation rate, which were implemented as human-specific parameters identical for all patients. These non-fixed parameters were re-estimated using the maximum likelihood method. The resulting parameter estimates are summarized in Dataset EV6.

## Correlation of model-based characteristics and patient-specific clinical features

The calibrated human model was used to simulate patient-specific characteristics such as the basal MET phosphorylation rate, the AUC of ppAKT, the total MET abundance and respective ratios. These characteristics were correlated to patient-specific clinical features calculating the Spearman correlation coefficients and respective $p$ values. In addition, partial Spearman correlations were calculated for a subset of patient features correcting for the confounding factors age, BMI, fibrosis and CCI (Kim, 2015).

# Data availability

The mass spectrometry proteomics data have been deposited to the ProteomeXchange Consortium through the PRIDE partner repository (Vizcaino et al, 2014) with the data set identifier PXD043007 (human) and PXD041563 (mouse). The ODE models developed for the mouse and the human setting are available on BioModels (Malik-Sheriff et al, 2020) in SBML and PEtab (Schmiester et al, 2021) formats under the identifier MODEL2306280002. The Source data of the figures is available on BioStudies.

## Peer review information

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

## Acknowledgements

The authors thank the Proteomics Core Facility of the German Cancer Research Center (DKFZ), especially Dominic Helm and Luisa E. Schwarzmüller, for providing support in establishing the DIA workflow and data analysis. Furthermore, we would like to thank Dirk Grimm for production of AAV particles and Marcus Rosenblatt for fruitful discussions during the development of the mathematical model. We thank Katharina Belgasmi, Lena Vlasov, Sandra Bonefas, and Alexander Held for technical assistance. We thank the Nikon Imaging Center (Heidelberg University) for providing access to their facility. This work was supported by the German Ministry of Education and Research (BMBF) within the LiSyM network [031L0042, 031L0045, 031L0048, 031L0049, 031L0052] and the LiSyM-Cancer networks SMART-NAFLD [031L0256A, 031L0256B, 031L0256C, 031L0256G], C-TIP-HCC [031L0257C, 031L0257D, 031L0257K], DEEP-HCC [031L0258E], the German Research Foundation (DFG) [457840828], the MSCoreSys network SMART-CARE [031L0212B] and by the German Center for Lung Research (DZL) [82DZL004A4]. The authors acknowledge support by the Open Access Publication Fund of the University of Freiburg and the state of Baden-Württemberg through bwHPC.

## Author contributions

**Sebastian Burbano de Lara**: Conceptualization; Investigation; Visualization; Writing—original draft. **Svenja Kemmer**: Conceptualization; Data curation; Formal analysis; Visualization; Methodology; Writing—original draft. **Ina Biermayer**: Conceptualization; Formal analysis; Investigation; Visualization; Writing—original draft. **Svenja Feiler**: Investigation. **Artyom Vlasov**: Formal analysis; Investigation. **Lorenza A D'Alessandro**: Conceptualization. **Barbara Helm**: Methodology. **Christina Mölders**: Investigation. **Yannik Dieter**: Investigation. **Ahmed Ghallab**: Resources. **Jan G Hengstler**: Resources. **Christiane Körner**: Investigation. **Madlen Matz-Soja**: Resources. **Christina Götz**: Investigation. **Georg Damm**: Resources. **Katrin Hoffmann**: Resources. **Daniel Seehofer**: Resources. **Thomas Berg**: Resources.

**Marcel Schilling**: Conceptualization; Supervision; Writing—original draft. **Jens Timmer**: Conceptualization; Supervision; Funding acquisition; Writing—review and editing. **Ursula Klingmüller**: Conceptualization; Supervision; Funding acquisition; Project administration; Writing—review and editing.

## Funding

## Disclosure and competing interests statement

The authors declare no competing interests. UK is an editorial advisory board member. This has no bearing on the editorial consideration of this article for publication.

# Expanded View Figures

**Figure EV1. Quantitative data for calibration of the mouse model.**

(A) Model calibration with HGF dose-resolved signal transduction measurements in primary mouse hepatocytes of SD and WD mice. Cells were stimulated with indicated doses of HGF for 10 min and phosphorylation of MET, ERK and AKT was quantified by immunoblotting. Signal is shown in arbitrary units (a.u.). Data points are displayed as dots with error bars representing $1\sigma$ confidence interval estimated from biological replicates ($n = 3$–9 per diet and dose) using a combined scaling and error model. Model trajectories are represented by solid lines. (B) Model calibration with HGF time-resolved signal transduction measurements in primary mouse hepatocytes of SD and WD mice. Immunoblot measurements for ERK, AKT and S6 abundance upon stimulation with 40 ng/ml HGF. Data points are displayed as dots along with error bars representing $1\sigma$ confidence interval estimated from biological replicates ($n = 3$–9 per diet and time point) using a combined scaling and error model. Model trajectories are depicted as solid lines. (C) A Bayesian information criterion (BIC) analysis was performed to determine the diet-specific parameters needed to describe the experimental data. The threshold for rejection was set to $\Delta\text{BIC} = 10$ as suggested (Lorah and Womack, 2019). H0 including 64 parameters could be reduced to H2.1.2 including 61 parameters, suggesting that only the basal phosphorylation rate of the HGF receptor MET was dysregulated between diets. (D) Correlation of basal MET phosphorylation and MET abundance at time point 0 h. Dots display the respective values for each mouse ($n = 9$ per diet). Correlation coefficient and $p$ value were calculated using a simple linear regression ($p$ value $= 0.19$).

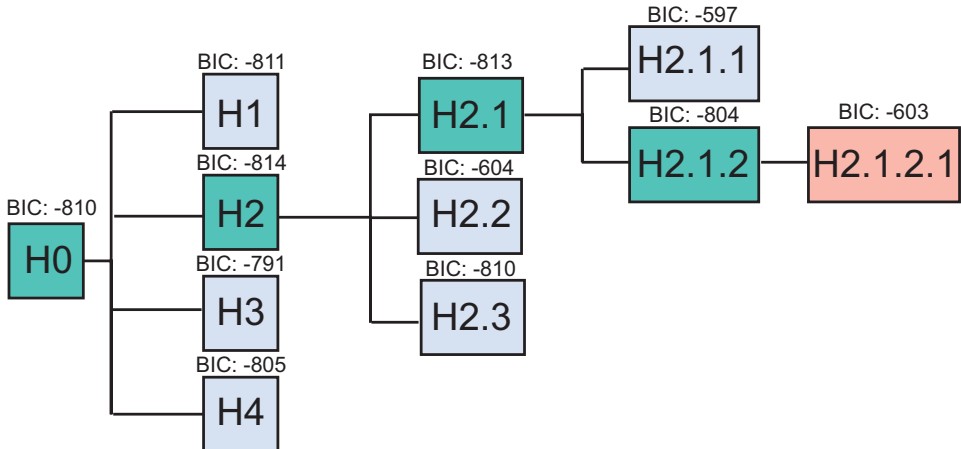

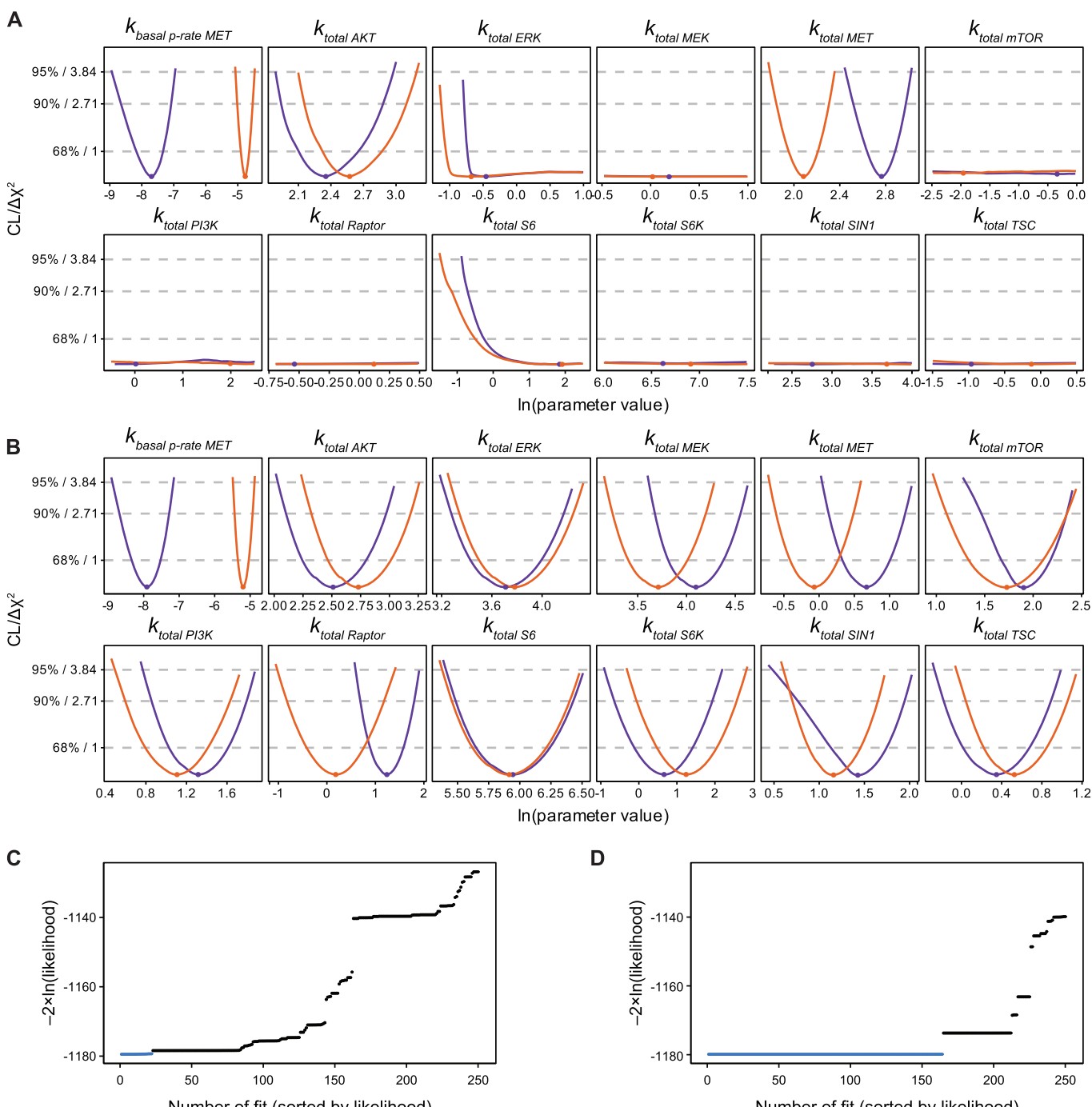

**Figure EV2. Impact of the DIA data on identifiability and convergence.**

(A) The profile likelihood as a measure of parameter identifiability (Raue et al, 2009) is depicted for all dysregulated parameters before implementation of the DIA data. If the negative log likelihood reaches a statistical threshold in both directions, the parameter has defined confidence bounds and is therefore called identifiable. If this limit is not reached on both sides, the parameter is classified as unidentifiable. Solid lines indicate the profile likelihood of dysregulated parameters for SD (purple) and WD (orange) along with the optimal parameter values as dots. Dashed lines depict thresholds for the confidence interval assessment. (B) The profile likelihood as a measure of parameter identifiability is depicted for all dysregulated parameters after implementation of the DIA data. (C) The convergence of the optimization before implementation of the DIA data is assessed based on a waterfall plot (Raue et al, 2013). This plot depicts the results of 250 optimization runs starting from randomly selected parameter sets sorted by the negative log likelihood. The global optimum, indicated in blue, was reached in 22 of the 250 cases. (D) The convergence of the optimization after implementation of the DIA data is assessed based on a waterfall plot. The global optimum, indicated in blue, was reached in 164 out of 250 cases.

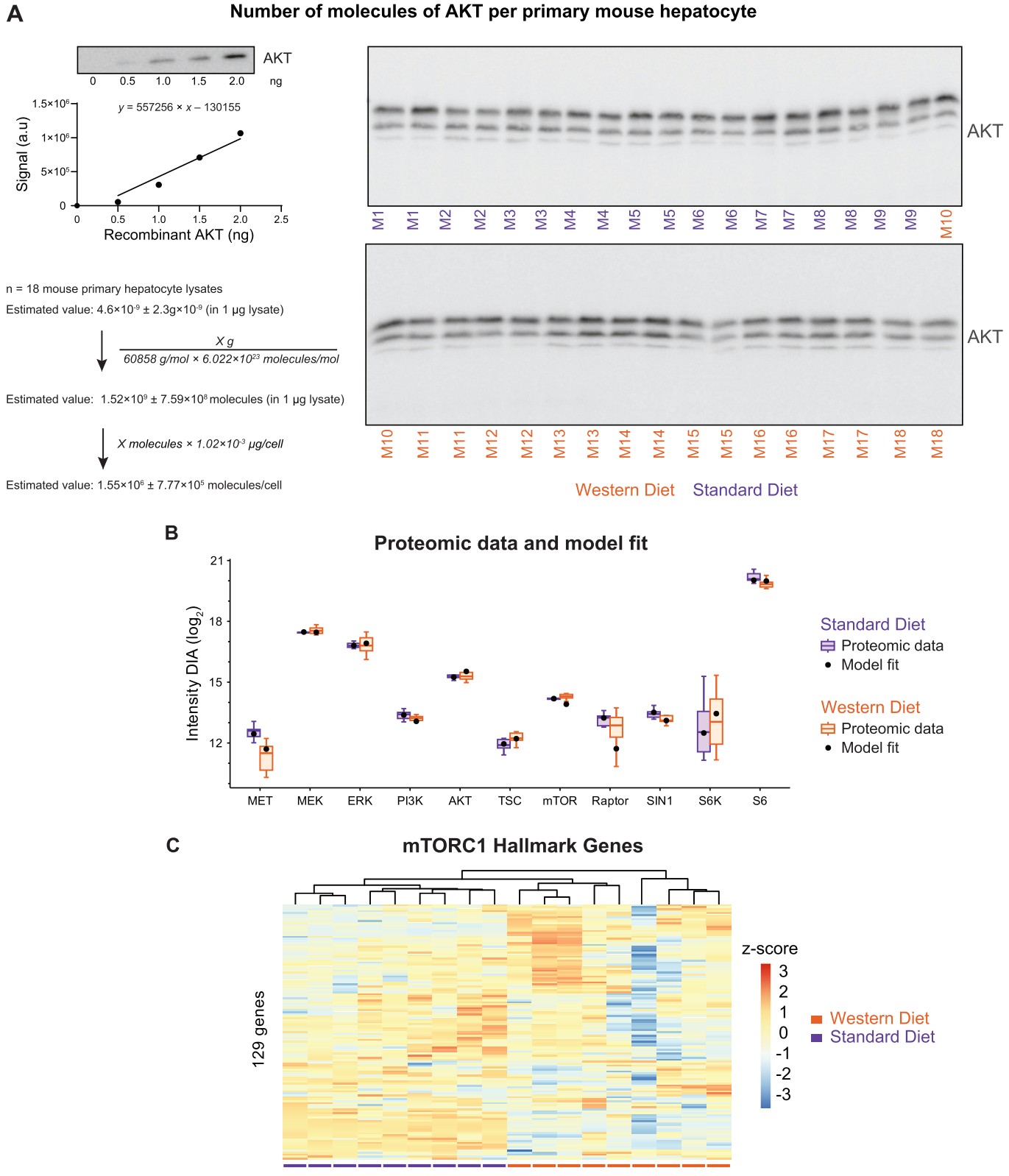

**A**

**Number of molecules of AKT per primary mouse hepatocyte**

$y = 557256 \times x - 130155$

n = 18 mouse primary hepatocyte lysates

Estimated value: $4.6 \times 10^{-9} \pm 2.3 \mathrm{g} \times 10^{-9}$ (in 1 µg lysate)

$$\frac{X\,g}{60858\ \mathrm{g/mol} \times 6.022 \times 10^{23}\ \mathrm{molecules/mol}}$$

Estimated value: $1.52 \times 10^{9} \pm 7.59 \times 10^{8}$ molecules (in 1 µg lysate)

*X molecules* $\times 1.02 \times 10^{-3}$ µg/cell

Estimated value: $1.55 \times 10^{6} \pm 7.77 \times 10^{5}$ molecules/cell

Western Diet    Standard Diet

**B**  **Proteomic data and model fit**

Standard Diet
⊟ Proteomic data
● Model fit

Western Diet
⊟ Proteomic data
● Model fit

**C**  **mTORC1 Hallmark Genes**

129 genes

18 samples

z-score

Western Diet
Standard Diet

◀ **Figure EV3. Absolute quantification of AKT and model-based estimations of total protein abundance.**

(A) Absolute number of molecules of AKT per primary mouse hepatocyte was determined by quantitative immunoblotting. Based on a dilution curve of recombinant AKT, the number of molecules of AKT in 1 μg lysate was determined. This value was converted with the total protein content per primary mouse hepatocyte into the number of molecules of AKT per cell. (B) Measurements of protein abundances derived from primary mouse hepatocytes were implemented in model calibration. Lysates of unstimulated hepatocytes were subjected to data-independent mass spectrometry analysis. Resulting data was LFQ normalized and represented as boxplot: center line indicates median; box limits indicate 25th to 75th percentiles. The lower and upper whiskers extend from the hinge to the smallest or largest value at most 1.5× interquartile range of the hinge. Dots represent the model fit ($n = 9$ per diet). (C) A list of Hallmark mTORC1 signaling genes was downloaded from Gene Set Enrichment Analysis (GSEA) and used to filter full proteomes of SD and WD mice. Out of 200 listed proteins, 129 were quantified in all samples and used to cluster samples based on protein abundance using the R package pheatmap.

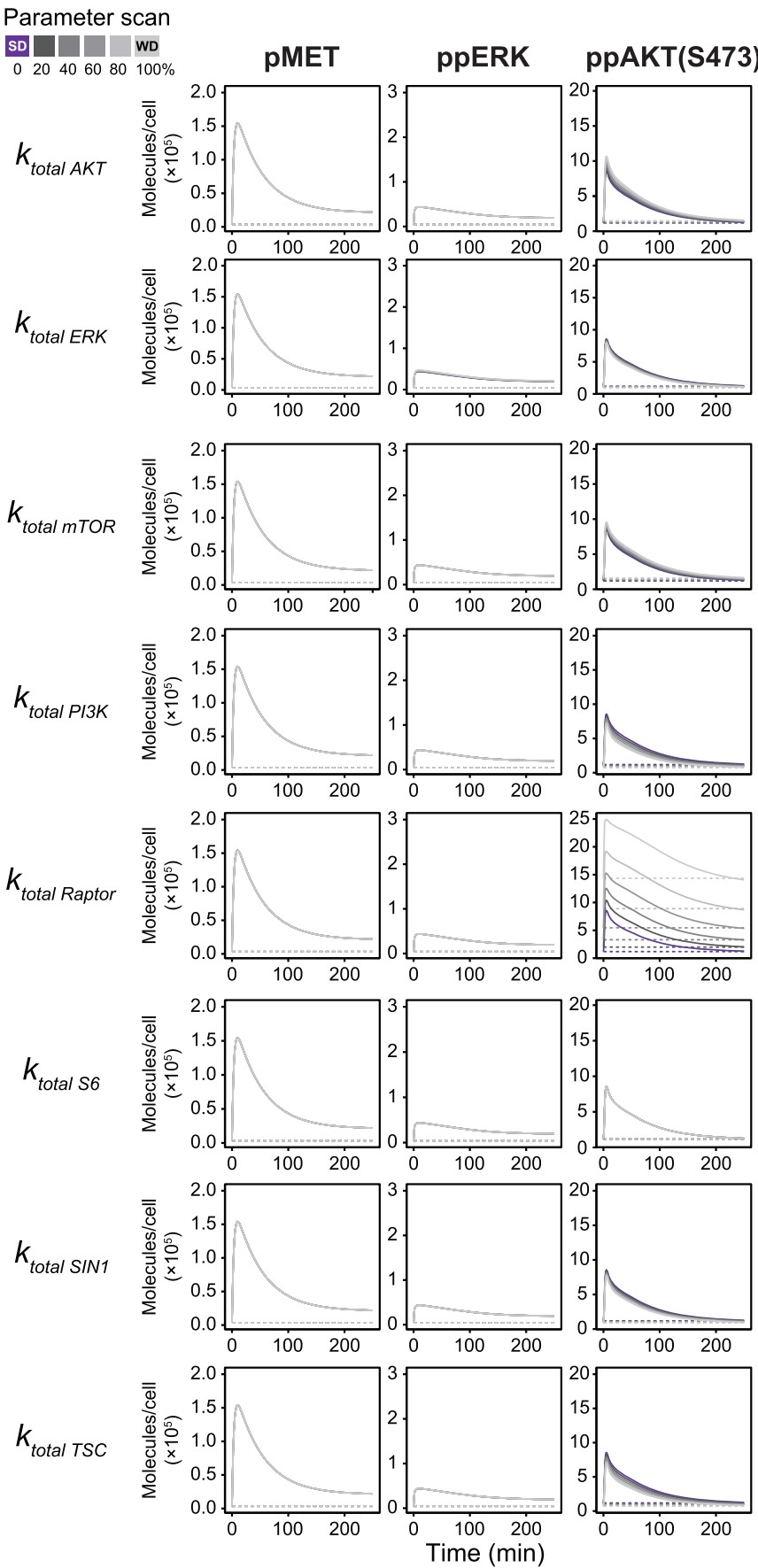

◄   **Figure EV4.   Influence of dysregulated parameters on protein dynamics.**

Individual parameter scan of one dysregulated parameter at a time as explained in Fig. 4A. The value for the indicated parameter was gradually shifted from the SD estimate (purple) to the WD estimate (light gray). The model simulations for the phosphorylation dynamics of MET, ERK and AKT are displayed in molecules/cell. Solid lines indicate model trajectories after HGF stimulation and dashed lines indicate basal levels.

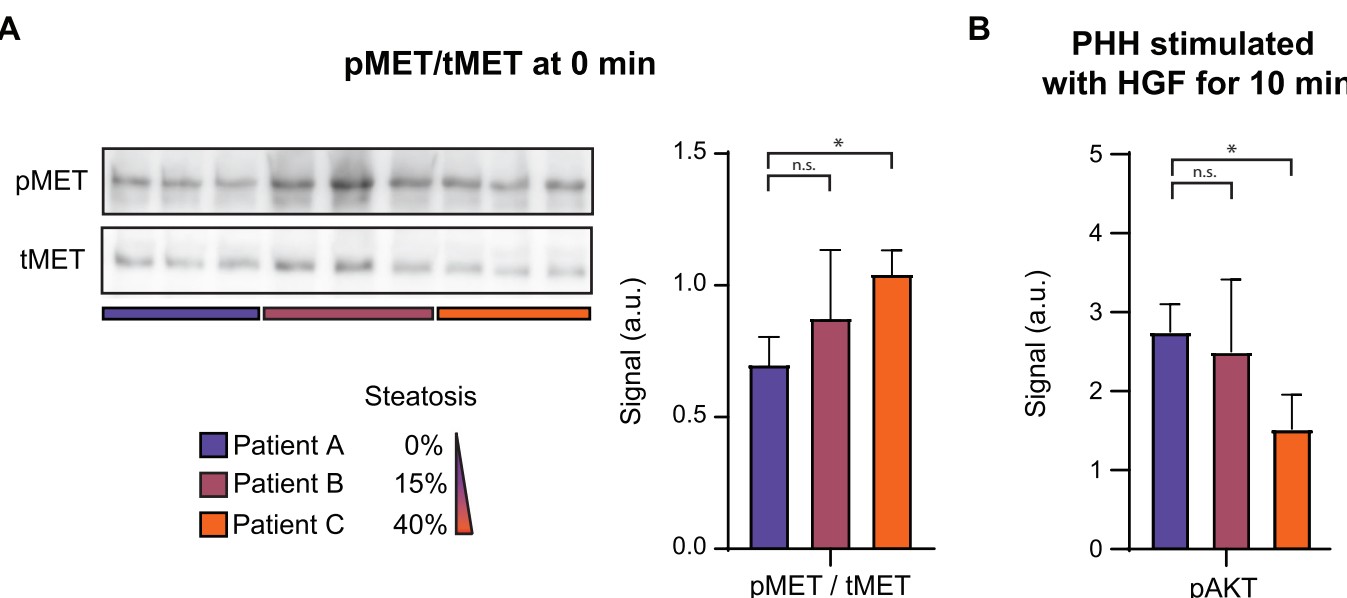

**Figure EV5. Signal transduction in primary human hepatocytes from steatotic patients.**

(A) Isolated primary human hepatocytes from patients with different levels of steatosis were analyzed using quantitative immunoblotting. The ratio of basal MET phosphorylation to MET abundance without HGF stimulation was quantified. Error bars represent one standard deviation ($n = 3$). (B) Primary human hepatocytes from patients with different levels of steatosis were stimulated with 40 ng/ml HGF. Phosphorylation of AKT was quantified by immunoblotting after 10 min. $p$ values were calculated using a two-tailed $t$ test (pMET/tMET *0.011, AKT *0.018). Error bars represent one standard deviation ($n = 3$).

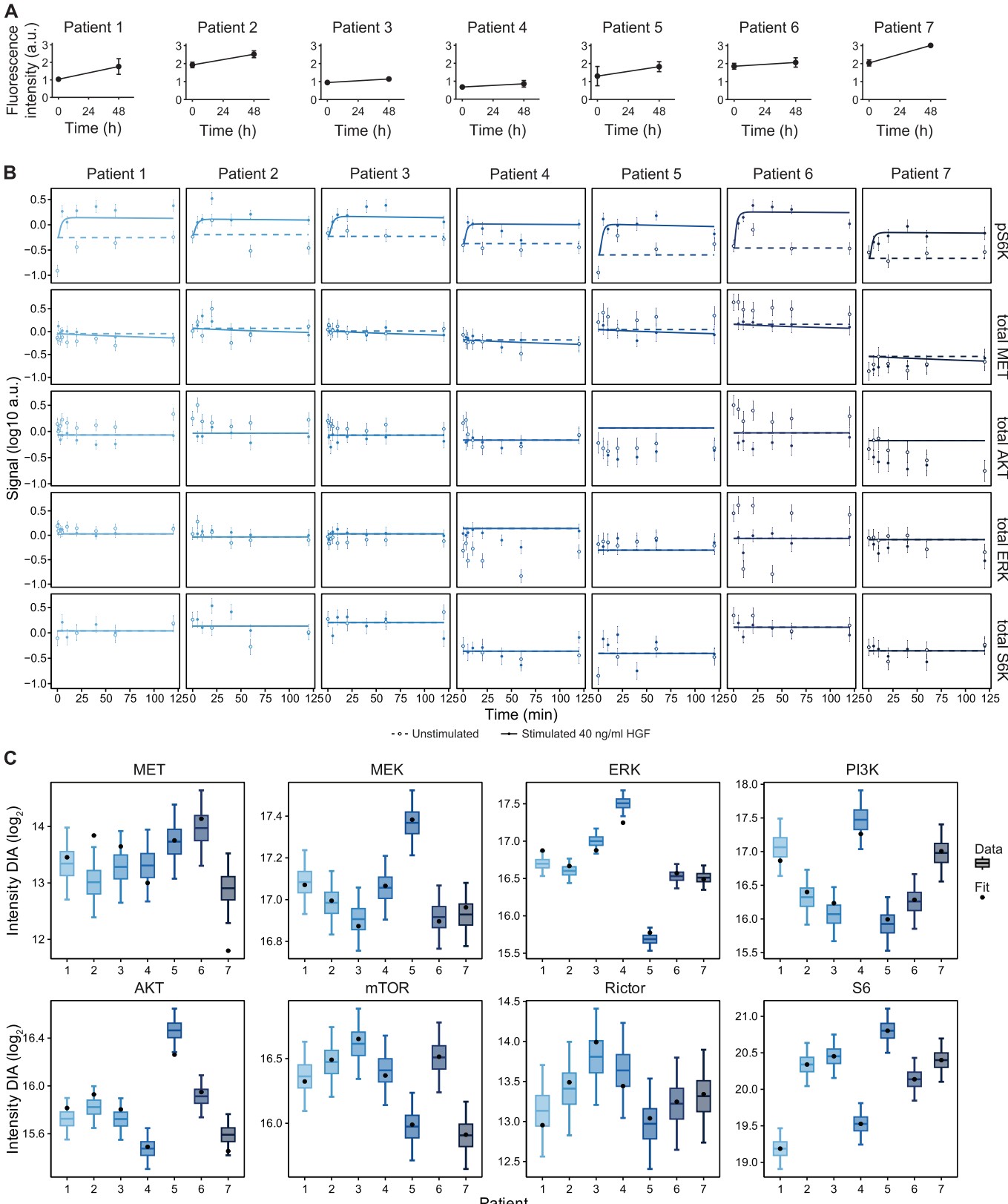

◀  **Figure EV6.  Quantitative data for calibration of the human model.**

(A) Proliferation measurements in isolated primary human hepatocytes from patients. Cells were stimulated with 40 ng/ml HGF. DNA content was measured at time point 0 h and after 48 h by staining with SYBRGreen I. (B) Time-resolved immunoblot measurements and model fits for pS6K as well as MET, AKT, ERK and S6K abundance in primary human hepatocytes derived from seven patients. Cells were stimulated with 40 ng/ml HGF or left untreated. Signal is shown in $\log_{10}$ arbitrary units (a.u.). Data points are displayed as dots along with error bars representing 1σ confidence interval estimated from technical replicates ($n = 1$–3 per patient) using a combined scaling and error model. Model trajectories are depicted as lines. (C) Measurements for protein abundances derived from primary patient hepatocytes were included as additional data for model calibration. Lysates of unstimulated hepatocytes were subjected to data-independent mass spectrometry analysis ($n = 1$–3 per patient). Resulting data was normalized using label-free quantification and represented as boxplot: center line indicates the patient median; box limits are defined as 1σ, calculated based on the mean spread of the cohort per protein. The lower and upper whiskers extend from the center line by 3σ. Model fits are represented as black dots.

