## [Peer Review File · Molecular Systems Biology]

Basal MET Phosphorylation is an Indicator of Hepatocyte Dysregulation in Liver Disease

Sebastian Burbano de Lara, Svenja Kemmer, Ina Biermayer, Svenja Feiler, Artyom Vlasov, Lorenza D'Alessandro, Barbara Helm, Christina Mölders, Yannik Dieter, Ahmed Ghallab, Jan G Hengstler, Christiane Körner, Madlen Matz-Soja, Christina Götz, Georg Damm, Katrin Hoffmann, Daniel Seehofer, Thomas Berg, Marcel Schilling, Jens Timmer, and Ursula Klingmueller
DOI: 10.15252/msb.202311856

Corresponding author(s): Ursula Klingmueller (u.klingmueller@dkfz.de), Jens Timmer (jeti@fdm.uni-freiburg.de), Marcel Schilling (m.schilling@dkfz.de)

Review Timeline:

Submission Date:	4th Jul 23
Editorial Decision:	11th Aug 23
Revision Received:	7th Nov 23
Editorial Decision:	30th Nov 23
Revision Received:	6th Dec 23
Accepted:	8th Dec 23

Editor: Maria Polychronidou

Transaction Report:

11th Aug 2023

Manuscript Number: MSB-2023-11856

Title: Basal MET Phosphorylation is an Indicator of Hepatocyte Dysregulation in Liver Disease

Dear Ursula,

Thank you again for submitting your work to Molecular Systems Biology. We have now heard back from the three reviewers who agreed to evaluate your study. As you will see below the reviewers appreciate the carefully performed analyses and think that the study seems relevant for a broad audience. They do however raise a series of concerns which we would ask you to address in a revision.

I think that the reviewers' comments seem rather clear and I therefore do not see the need to repeat any of them here. As you will see below, reviewer #1 recommends performing follow up validations in vivo or ex vivo. While we would encourage you to include such data if they are already available or if you think it seems reasonable to obtain them within the timeframe of a major revision, we do not think that these analyses are mandatory for the acceptance of the study. All other issues raised by the reviewers need to be satisfactorily addressed. As you may already know, our editorial policy allows in principle a single round of major revision, so it is essential to provide responses to the reviewers' comments that are as complete as possible. Please feel free to contact me in case you would like to discuss in further detail any of the issues raised. If helpful, I would be happy to schedule a call.

On a more editorial level, we would ask you to address the following points:

- We have replaced Supplementary Information by the Expanded View (EV format). In this case, all additional figures* can be provided as EV Figures. Please provide one file per EV Figure. Their legends should be included in the manuscript text. *If after the performed revisions the number of additional Figures is > 6, then they can all be included in a PDF called Appendix. Appendix figures should be labeled and called out as: "Appendix Figure S1, Appendix Figure S2... Appendix Table S1..." etc. Each legend should be below the corresponding Figure/Table in the Appendix. Please include a Table of Contents in the beginning of the Appendix. For detailed instructions regarding expanded view please refer to our Author Guidelines: .

- For the EV Datasets: please include a description of each dataset, either as a separate sheet (for Excel files) or as a README.txt file zipped together with the EV Dataset (for .csv files).

- Please provide a "standfirst text" summarizing the study in one or two sentences (approximately 250 characters), three to four "bullet points" highlighting the main findings and a "synopsis image" (550px width and max 400px height, jpeg format) to highlight the paper on our homepage.

- All Materials and Methods need to be described in the main text. We would encourage you to use 'Structured Methods', our new Materials and Methods format. According to this format, the Materials and Methods section should include a Reagents and Tools Table (listing key reagents, experimental models, software and relevant equipment and including their sources and relevant identifiers) followed by a Methods and Protocols section in which we encourage the authors to describe their methods using a step-by-step protocol format with bullet points, to facilitate the adoption of the methodologies across labs. More information on how to adhere to this format as well as downloadable templates (.doc or .xls) for the Reagents and Tools Table can be found in our author guidelines: . An example of a Method paper with Structured Methods can be found here:

- For data quantification: please specify the name of the statistical test used to generate error bars and P values, the number (n) of independent experiments (specify technical or biological replicates) underlying each data point and the test used to calculate p-values in each figure legend. The figure legends should contain a basic description of n, P and the test applied. Graphs must include a description of the bars and the error bars (s.d., s.e.m.).

- When you resubmit your manuscript, please download our CHECKLIST (<https://bit.ly/EMBOPressAuthorChecklist>) and include the completed form in your submission.

Please note that the Author Checklist will be published alongside the paper as part of the transparent process (<https://www.embopress.org/page/journal/17444292/authorguide#transparentprocess>).

If you feel you can satisfactorily deal with these points and those listed by the referees, you may wish to submit a revised version of your manuscript. Please attach a covering letter giving details of the way in which you have handled each of the points raised by the referees. A revised manuscript will be once again subject to review and you probably understand that we can give you no guarantee at this stage that the eventual outcome will be favorable.

Best wishes,

Maria

Maria Polychronidou, PhD
Senior Editor
Molecular Systems Biology

We realize that it is difficult to revise to a specific deadline. In the interest of protecting the conceptual advance provided by the work, we recommend a revision within 3 months (9th Nov 2023). Please discuss the revision progress ahead of this time with the editor if you require more time to complete the revisions. Use the link below to submit your revision:

IMPORTANT: When you send your revision, we will require the following items:

1. the manuscript text in LaTeX, RTF or MS Word format
2. a letter with a detailed description of the changes made in response to the referees. Please specify clearly the exact places in the text (pages and paragraphs) where each change has been made in response to each specific comment given
3. three to four 'bullet points' highlighting the main findings of your study
4. a short 'blurb' text summarizing in two sentences the study (max. 250 characters)
5. a 'thumbnail image' (550px width and max 400px height, Illustrator, PowerPoint or jpeg format), which can be used as 'visual title' for the synopsis section of your paper.
6. Please include an author contributions statement after the Acknowledgements section (see <https://www.embopress.org/page/journal/17444292/authorguide>)
7. Please complete the CHECKLIST available at (<https://bit.ly/EMBOPressAuthorChecklist>). Please note that the Author Checklist will be published alongside the paper as part of the transparent process (<https://www.embopress.org/page/journal/17444292/authorguide#transparentprocess>).
8. When assembling figures, please refer to our figure preparation guideline in order to ensure proper formatting and readability in print as well as on screen:
<https://bit.ly/EMBOPressFigurePreparationGuideline>
See also figure legend guidelines: <https://www.embopress.org/page/journal/17444292/authorguide#figureformat>
9. Please note that corresponding authors are required to supply an ORCID ID for their name upon submission of a revised manuscript (EMBO Press signed a joint statement to encourage ORCID adoption). (<https://www.embopress.org/page/journal/17444292/authorguide#editorialprocess>)
Currently, our records indicate that the ORCID for your account is 0000-0001-9845-3099.

Link Not Available

The system will prompt you to fill in your funding and payment information. This will allow Wiley to send you a quote for the article processing charge (APC) in case of acceptance. This quote takes into account any reduction or fee waivers that you may be eligible for. Authors do not need to pay any fees before their manuscript is accepted and transferred to the publisher.

EMBO Press participates in many Publish and Read agreements that allow authors to publish Open Access with reduced/no publication charges. Check your eligibility: <https://authorservices.wiley.com/author-resources/Journal-Authors/open-access/affiliation-policies-payments/index.html>

*** PLEASE NOTE *** As part of the EMBO Press transparent editorial process initiative (see our Editorial at <https://dx.doi.org/10.1038/msb.2010.72>), Molecular Systems Biology publishes online a Review Process File with each accepted manuscripts. This file will be published in conjunction with your paper and will include the anonymous referee reports, your point-by-point response and all pertinent correspondence relating to the manuscript. If you do NOT want this File to be published, please inform the editorial office at msb@embo.org within 14 days upon receipt of the present letter.

Reviewer #1:

summary

In this manuscript de Lara and colleagues study signalling dysregulation in the context of liver disease. They combine quantitative immunoblotting, proteomics and live cell imaging with dynamic modelling to identify basal MET phosphorylation as key indicator of signalling regulation. First, they establish the role of basal MET phosphorylation as explanatory variable for diet-induced changes to MAPK and PI3K signalling in murine model, then confirm the role of these pathways in proliferation control. They then go on to analyse patient data and propose basal MET phosphorylation as predictive variable for post-operational response.

general remarks

Given the previously established role of MET in liver regeneration, these findings are not necessarily surprising from a biological perspective but therefore also intuitively plausible. In contrast, given the lack of prior pre-operative predictor for surgery outcome, these findings appear to be relevant from a clinical perspective. The combination of murine human models; state-of-the-art experimental techniques and dynamic modelling strategies; as well as the excellent writing and good text flow make for a compelling story that should be interesting for a diverse audience including modellers, biologists as well as clinicians. It was a pleasure to review this manuscript and I am convinced this is a strong manuscript suitable for publication in MSB.

major points

- Even though the use of proteomic data to explain differences between between diets and patients appears attractive, I am not entirely convinced that this is really necessary and might in fact weaken the importance of basal MET phosphorylation. This impression is primarily based on the overlapping confidence intervals/profiles in figure EV2 as well as the low (albeit sometimes statistically significant) inter-diet and inter-patient variability in protein expression. This is relevant given the role of MET phosphorylation in the regulation of MET degradation and thus it's impact on the total abundance of MET. Accordingly, I find it plausible that basal phosphorylation might explain all variability in MET expression, in which case MET expression levels could be an attractive clinical variable as indirect indicator of basal MET phosphorylation. This might be easier to quantify experimentally and could already be tested with the available data. I don't think this investigation would be essential to support the key conclusions of the manuscript, but has the potential to strengthen the manuscript by bolstering the importance of MET phosphorylation with relatively little time investment.

minor points

- given that experiments were performed under starvation conditions, I find the observation of basal MET activation surprising. the authors may want to speculate in the discussion what the mechanism of activation is? Is there any literature evidence for incomplete auto-inhibition, ligand-independent activation of the receptor or sporadic pulsatile activation through paracrine mechanisms?
- I had difficulties following and understanding the relevance of the section "The basal phosphorylation rate parameter of MET, ..." so the authors may want to consider removing or rephrasing the respective section.
- the most striking difference between patient vs murine data appears to be transient vs sustained activation of signalling, maybe the authors want to comment on that
- I had some difficulties understanding figure 6B, maybe the relevant passages in the text and figure legends could be rephrased? e.g. what does dot size indicate? Does it make sense to look at partial correlations here?
- some of the supporting information would benefit from more metadata/description and/or cross checking with the manuscript. For example in EV4, why is $\exp(\text{MET_SD})$ used as transformation for both MET_SD and MET_WD? Shouldn't there be $\exp(\text{MET_SD})$ and $\exp(\text{MET_WD})$. Does $k_{\text{MET_expression_SD}}$ correspond to $k_{\text{total_MET}}$?

Reviewer #2:

In the manuscript entitled "Basal MET Phosphorylation is an Indicator of Hepatocyte Dysregulation in Liver Disease" Burbano de Lara and colleagues employ a murine model for steatotic liver disease combined with proteomic analysis of primary hepatocytes from these animals and dynamic pathway modeling to determine metabolic and signaling changes induced by steatosis. In particular, in mice on WD baseline MET phosphorylation is increased, while the responsiveness to HGF is reduced. This has implications on proliferative capacity of hepatocytes, as impressively demonstrated in human tissue samples.

Overall, this is a very comprehensive study supported by robust mathematical modeling. This is an important study to highlight the altered responsiveness to proliferative growth signals in patients with metabolic signaling.

One substantial limitation of the study is the focus on disaggregated primary hepatocytes. Given the importance of hepatic

zonation with regard to both metabolic capacity, susceptibility to metabolic insult (zone 3 is predominantly affected by steatosis), it would be important to highlight these aspects. While it would be unreasonable to expect these studies to be supplemented by rigorous tissue-level analyses, these aspects should at least be discussed.

It would be important to highlight reduced regenerative capacity by complementary analysis to FUCCI transgenic lines, such as BrdU incorporation or PCNA staining.

In light of recent nomenclature changes, the authors should no longer refer to NAFLD, but at least once also highlight that this disease is no being relabeled as metabolic-dysfunction associated steatotic liver disease (MASLD):
https://easl.eu/news/new_fatty_liver_disease_nomenclature-2/

Reviewer #3:

This manuscript by Burbano de Lara et. al. uses global proteomics and dynamic pathway modeling in mouse and human primary hepatocytes to study hepatocyte dysregulation in WD feeding and NAFLD. The authors constructed and calibrated a dynamic pathway model for HGF signal transduction using data from murine hepatocytes, and then adapted this model to data from patient-derived primary human hepatocytes. Authors describe that HGF-signal transduction is strikingly altered in WD hepatocytes with a significant reduction in AKT phosphorylation. Further, they propose that the change in basal MET phosphorylation rate parameter sufficiently describes altered proliferative HGF signaling in WD hepatocytes. The authors also demonstrate that levels of basal MET phosphorylation are correlated well with patient outcomes and suggest its use in predicting liver disease burden.

The manuscript is interesting, and the observations are presented appropriately. Particularly, the study sheds light on how a major proliferative program in the liver is disrupted in NAFLD. The major drawback of this study is that molecular insights obtained from dynamic modeling are not validated with appropriate in-vivo or ex-vivo experimentation.

Major comments:

1. The study proposes a central role for changes in basal MET phosphorylation in driving WD-specific alterations in HGF-induced signal transduction. This fundamental finding must be validated with in-vivo or ex-vivo (primary hepatocyte-based) experiments.
2. The involvement of mTOR pathway in regulating AKT phosphorylation in WD hepatocytes is highlighted in the study, but only in the mathematical models. Experimental evidence from primary hepatocytes must be presented to validate the assumptions/conclusions.
3. It would be appropriate to show that ribosomal protein S6 phosphorylation is altered in conditions under evaluation using western blotting etc.
4. Authors show that there is a marked increase in cell cycle entries and proliferation rate in WD compared to SD hepatocytes. Evidence must be presented to support that pMET levels in fact drive (or contribute significantly to) this. Current experimental data does not show this.
5. Authors have used quantitative western blotting to estimate parameters used in the study, but none of the blots are presented in the paper/made available as supplementary. Immunoblot-based quantifications of the pMET level in the patient-derived hepatocytes are not presented either. All western blotting images must be made available. Also, catalog numbers of antibodies, etc. must be provided to enable reproducibility.
6. Recovery experiments in WD hepatocytes by perturbing levels of pMET, phospho-AKT, mTOR components etc. would be appropriate for substantiating authors' claims about regulatory mechanisms of AKT phosphorylation and hepatocyte proliferation in WD/SD.

Minor comments:

7. Does Fig 3B represent an average of N=9 mice? If so, plot data from each mouse, as individual data points overlaid on the current averaged box plot.
8. In discussion, line 431, change "proliferation of hepatocytes" to "ex-vivo proliferation of primary hepatocytes". The authors must highlight that observations made in the study are ex-vivo and not in-vivo.
9. Line 330-332 repeat immediately after that.
10. Fig. EV3A shows the immunoblot used for generating a standard curve using recombinant AKT. The western blots used for AKT concentration assessment should also be included.

In the following we provide a point-by-point reply to the comments of the reviewers. All changes of the manuscript are described in this reply letter citing the respective page, line and specific passage of the main text. Additionally, changes are highlighted in yellow in both documents. The DOI of publications only cited in this reply letter, but not in the main manuscript, is provided in the text.

Reviewer #1:

##summary

In this manuscript de Lara and colleagues study signalling dysregulation in the context of liver disease. They combine quantitative immunoblotting, proteomics and live cell imaging with dynamic modelling to identify basal MET phosphorylation as key indicator of signalling regulation. First, they establish the role of basal MET phosphorylation as explanatory variable for diet-induced changes to MAPK and PI3K signalling in murine model, then confirm the role of these pathways in proliferation control. They then go on to analyse patient data and propose basal MET phosphorylation as predictive variable for post-operational response.

##general remarks

Given the previously established role of MET in liver regeneration, these findings are not necessarily surprising from a biological perspective but therefore also intuitively plausible. In contrast, given the lack of prior pre-operative predictor for surgery outcome, these findings appear to be relevant from a clinical perspective. The combination of murine human models; state-of-the-art experimental techniques and dynamic modelling strategies; as well as the excellent writing and good text flow make for a compelling story that should be interesting for a diverse audience including modellers, biologists as well as clinicians. It was a pleasure to review this manuscript and I am convinced this is a strong manuscript suitable for publication in MSB.

##major points

- Even though the use of proteomic data to explain differences between diets and patients appears attractive, I am not entirely convinced that this is really necessary and might in fact weaken the importance of basal MET phosphorylation. This impression is primarily based on the overlapping confidence intervals/profiles in figure EV2 as well as the low (albeit sometimes statistically significant) inter-diet and inter-patient variability in protein expression.

We thank the reviewer for this comment. To clarify the point raised by the reviewer, we extended Fig EV3A and improved the explanation of the importance of the proteomics data to increase parameter identifiability (main text, line 226-245):

Based on previous experience (Adlung et al., 2017), we reasoned that inclusion of the basal abundance of all key protein species would improve parameter identifiability. Therefore, total protein lysates of SD and WD primary hepatocytes were examined by global proteomics employing mass spectrometric analysis operated in the data independent acquisition (DIA) mode that facilitated reproducible coverage and reliable quantification. These determinations revealed a significant decrease in the intensity of MET, SIN1 and S6 and a significant increase of TSC in the WD primary hepatocytes, while the intensity of the other protein species was comparable between SD and WD primary hepatocytes, respectively (Fig 3B). Since the relative amount, which is based on the intensity determined by global proteomics, scales with the concentration of the examined proteins, we exemplarily determined the abundance of AKT in the SD and WD primary hepatocytes with quantitative immunoblotting (Fig EV3A). These quantifications and the knowledge of the average protein content of the primary hepatocytes allowed us to estimate the

number of AKT molecules per cell, which was included as additional information in the mathematical model. By linking the relative amount of AKT protein determined as intensity by our mass spectrometry-based DIA measurements with the corresponding absolute amount of AKT molecules per cell determined by quantitative immunoblotting, the model could infer the corresponding protein concentrations for all proteins of interest. With this additional information the mathematical model was able to estimate all twelve dysregulated parameters with narrow confidence intervals in both conditions (Fig EV2B).

This is relevant given the role of MET phosphorylation in the regulation of MET degradation and thus it's impact on the total abundance of MET. Accordingly, I find it plausible that basal phosphorylation might explain all variability in MET expression, in which case MET expression levels could be an attractive clinical variable as indirect indicator of basal MET phosphorylation. This might be easier to quantify experimentally and could already be tested with the available data. I don't think this investigation would be essential to support the key conclusions of the manuscript, but has the potential to strengthen the manuscript by bolstering the importance of MET phosphorylation with relatively little time investment.

We agree with the reviewer that MET expression levels could be a clinically easier accessible parameter as an indirect indicator of MET phosphorylation levels. Inspired by the suggestion of the reviewer we analyzed the correlation between levels of basal MET phosphorylation and total MET in SD and WD hepatocytes. The results displayed in the new figure (EV1D) unexpectedly revealed that there is no significant correlation between both. These observations were now included in the revised text (line 216-219):

To evaluate whether the abundance of total MET could serve as an indicator for the levels of phosphorylated MET, we examined their correlation. The results displayed in Figure EV1D showed that differences in the abundance of total MET do not allow to conclude on the levels of basal phosphorylated MET.

##minor points

- given that experiments were performed under starvation conditions, I find the observation of basal MET activation surprising. the authors may want to speculate in the discussion what the mechanism of activation is? Is there any literature evidence for incomplete auto-inhibition, ligand-independent activation of the receptor or sporadic pulsatile activation through paracrine mechanisms?

We followed the advice of the reviewer and included in the discussion of the revised manuscript in parts based on literature evidence hypotheses on a possible mechanisms of HGF-independent MET activation (main text, discussion, lines 449-454 and 483-485)

In this context, it is relevant to mention that others have presented evidence for a heterodimeric interaction between insulin receptor and MET (Fafalios *et al*, 2011), which could be involved in the HGF independent basal phosphorylation of MET in WD mice. This heterodimer and the known interactions between MET and the insulin receptor substrate 1 and 2 (IRS1 and IRS2) (Fafalios *et al.*, 2011) (DeAngelis *et al*, 2010) have not been explicitly included in our model, (...)

We have not directly addressed the molecular mechanism driving the increased basal phosphorylation of MET, but hypothesize that the diet-induced changes in the cell surface membrane of hepatocytes may alter the dimerization rate of MET receptors and change the heterodimerization capabilities of MET rendering the receptor a central integrator of metabolic changes.

- I had difficulties following and understanding the relevance of the section "The basal phosphorylation rate parameter of MET, ..." so the authors may want to consider removing or rephrasing the respective section.

To improve the understanding and to clarify the importance of this section, we updated Figure 4 and rephrased the respective section in the results part (277-308):

The basal phosphorylation rate parameter of MET is sufficient to explain altered HGF-signal transduction in WD hepatocytes

To investigate the individual impact of the twelve diet-specific parameters on WD-specific changes in the dynamics of HGF induced signal transduction, we utilized our mathematical model to perform a simulation analysis. As a starting point, all model parameters were set to the estimates for SD primary hepatocytes, which allowed to capture the dynamics of HGF signal transduction in these hepatocytes (Fig 4A, 1. Starting point). Subsequently, we gradually (20% intervals) changed only one dysregulated parameter at a time until it reached the parameter value estimated for WD hepatocytes (Fig 4A, 2. Simulation). We performed this analysis with all twelve parameters, which allowed us to assess the individual impact of each specific parameter by comparing the simulation results to the dynamics in WD primary hepatocytes (Fig 4A, 3. Comparison). Since we identified the increased basal phosphorylation of MET and ERK and the decreased area under the curve (AUC) of ppAKT upon HGF stimulation as main features differing between SD and WD primary hepatocytes, we simulated the trajectories of pMET, pERK and ppAKT (Thr 308, Ser 473). The model simulations (Fig 4B and Fig EV4) showed the largest effects when altering the basal protein abundance of MET ($k_{total\ MET}$), MEK ($k_{total\ MEK}$) and S6K ($k_{total\ S6K}$) as well as the basal phosphorylation rate of MET ($k_{basal\ p-rate\ MET}$). However, most of the parameters could not reproduce the features observed for WD primary hepatocytes. For example, a shift of $k_{total\ S6K}$ to the WD estimate resulted in a decrease in AKT phosphorylation but had no effect on the basal MET and ERK phosphorylation levels. Interestingly, when shifting the value of $k_{basal\ p-rate\ MET}$ to the estimated value for WD primary hepatocytes, all three features observed in WD primary hepatocytes were reproduced. Thus, only $k_{basal\ p-rate\ MET}$ was essential to reproduce all three features observed for WD primary hepatocytes.

To quantitatively assess how accurately these three features could be reproduced by the described parameter tuning, we performed an optimization analysis. During the analysis the mathematical model can choose an arbitrary value between the values in SD and WD for the analyzed parameter maximizing the reproducibility of the true WD features. In line with the results obtained by the simulation analysis, the only parameter that quantitatively reproduced the WD-specific features was the basal MET phosphorylation rate (Fig 4C), supporting our hypothesis of its key role as driver for the WD-specific alterations in HGF-induced signal transduction.

- the most striking difference between patient vs murine data appears to be transient vs sustained activation of signalling, maybe the authors want to comment on that

Please note that because the patient data showed a large heterogeneity between patient, the data was displayed in log scale and therefore appears more sustained compared to the mouse data. To address the point raised by the reviewer we include the following remark in the main text (line 386-388):

The patient data (displayed in log scale) was more heterogenous but overall resembled the dynamic behavior observed in the primary hepatocytes of the inbred mouse model.

- I had some difficulties understanding figure 6B, maybe the relevant passages in the text and figure legends could be rephrased? e.g. what does dot size indicate? Does it make sense to look at partial correlations here?

We thank the reviewer for this suggestion. To improve the understanding of Figure 6B we rephrased the text including figure legend and followed the advice to include a partial correlation (line 416-422):

Lastly, we corrected for patient specific features by performing a partial correlation to remove effects of confounding factors. After correction for the confounding factors age, BMI, fibrosis and CCI the correlation of patient outcome with proliferation and $k_{total\ MET}$ increased, albeit not to a significant extent. Importantly, the significant correlation between $k_{basal\ p-rate\ MET}$ and $basal\ p-rate\ MET / k_{total\ MET}$ with complication index and hospitalization days was maintained. Taken together our results suggested that $k_{basal\ p-rate\ MET}$ and $k_{basal\ p-rate\ MET} / k_{total\ MET}$ might provide as suitable measure to predict patient outcome.

- some of the supporting information would benefit from more metadata/description and/or cross checking with the manuscript. For example in EV4, why is $\exp(MET_SD)$ used as transformation for both MET_SD and MET_WD ? Shouldn't there be $\exp(MET_SD)$ and $\exp(MET_WD)$. Does $k_MET_expression_SD$ correspond to k_{total_MET} ?

We thank the reviewer for detecting this error in "Dataset_EV4_Parameter_transformations.csv". We corrected the dataset and included a README file for all Datasets following the guidelines suggested by MSB. In this README file we also clarified that indeed "k_MET_expression_SD"

and "k_MET_expression_WD" describes the differences of total MET abundance between WD and SD hepatocytes.

Reviewer #2:

In the manuscript entitled "Basal MET Phosphorylation is an Indicator of Hepatocyte Dysregulation in Liver Disease" Burbano de Lara and colleagues employ a murine model for steatotic liver disease combined with proteomic analysis of primary hepatocytes from these animals and dynamic pathway modeling to determine metabolic and signaling changes induced by steatosis. In particular, in mice on WD baseline MET phosphorylation is increased, while the responsiveness to HGF is reduced. This has implications on proliferative capacity of hepatocytes, as impressively demonstrated in human tissue samples.

Overall, this is a very comprehensive study supported by robust mathematical modeling. This is an important study to highlight the altered responsiveness to proliferative growth signals in patients with metabolic signaling.

One substantial limitation of the study is the focus on disaggregated primary hepatocytes. Given the importance of hepatic zonation with regard to both metabolic capacity, susceptibility to metabolic insult (zone 3 is predominantly affected by steatosis), it would be important to highlight these aspects. While it would be unreasonable to expect these studies to be supplemented by rigorous tissue-level analyses, these aspects should at least be discussed.

We thank the reviewer for pointing this out. We have now included a section in the discussion addressing the potential role of zonation in the liver (line 494-501):

The liver is organized in hexagonal liver lobules consisting of plates of hepatocytes interspersed by small capillaries, the liver sinusoids. Blood enters the hexagonal lobules through portal veins and drains towards a central vein, establishing a gradient of nutrients, metabolites and oxygen that correlates with distinct enzymatic equipment of hepatocytes and results in metabolic zonation of the liver lobule. As steatosis primarily affects the pericentral zone it is conceivable that susceptibility to metabolic insults might differ. We attempted to address the distribution of basal phosphorylated MET by histological stainings, but so far, the results remained inconclusive and require further optimization.

It would be important to highlight reduced regenerative capacity by complementary analysis to FUCCI transgenic lines, such as BrdU incorporation or PCNA staining.

To corroborate our observations obtained by life cell imaging of primary murine hepatocytes isolated from SD and WD mice showing that WD hepatocytes display increased HGF-independent proliferation, we now included in the new Figure 5 D the results of a SYBR Green assay detecting changes in the DNA content and adjusted the text as follows (line 330-335):

To corroborate these findings, we utilized the fluorescent dye SYBR Green, which binds to double stranded DNA, to quantify the DNA content in SD and WD primary hepatocytes left unstimulated or stimulated with HGF. In agreement with the results obtained with the FUCCI mouse derived primary hepatocytes, WD primary hepatocytes showed an increase in DNA content even in the absence of HGF, indicating that these primary hepatocytes were able to proliferate independently of HGF stimulation (Fig 5D).

In light of recent nomenclature changes, the authors should no longer refer to NAFLD, but at least once also highlight that this disease is no being relabeled as metabolic-dysfunction associated steatotic liver disease (MASLD): https://easl.eu/news/new_fatty_liver_disease_nomenclature-2/

We thank the reviewer for pointing this out, and included a reference to these changes in nomenclature in the introductory section of the main text (line 54-58):

In 2023 at the EASL Congress the nomenclature for NAFLD was updated to better reflect the improved understanding of the disease as metabolic dysfunction-associated liver disease (MASLD). These developments highlight that an understanding of the underlying mechanisms of disease development and progression is pivotal.

Reviewer #3:

This manuscript by Burbano de Lara et. al. uses global proteomics and dynamic pathway modeling in mouse and human primary hepatocytes to study hepatocyte dysregulation in WD feeding and NAFLD. The authors constructed and calibrated a dynamic pathway model for HGF signal transduction using data from murine hepatocytes, and then adapted this model to data from patient-derived primary human hepatocytes. Authors describe that HGF-signal transduction is strikingly altered in WD hepatocytes with a significant reduction in AKT phosphorylation. Further, they propose that the change in basal MET phosphorylation rate parameter sufficiently

describes altered proliferative HGF signaling in WD hepatocytes. The authors also demonstrate that levels of basal MET phosphorylation are correlated well with patient outcomes and suggest its use in predicting liver disease burden.

The manuscript is interesting, and the observations are presented appropriately. Particularly, the study sheds light on how a major proliferative program in the liver is disrupted in NAFLD. The major drawback of this study is that molecular insights obtained from dynamic modeling are not validated with appropriate in-vivo or ex-vivo experimentation.

Major comments:

1. The study proposes a central role for changes in basal MET phosphorylation in driving WD-specific alterations in HGF-induced signal transduction. This fundamental finding must be validated with in-vivo or ex-vivo (primary hepatocyte-based) experiments.

All experiment presented in our study were performed with primary hepatocytes from mice (wildtype or Fucci) exposed to standard or Western Diet and with primary hepatocytes from patients undergoing hepatectomy. To, as requested by the reviewer, further support our findings, we now included in a new Figure EV5 A and B the analysis of phosphorylated MET and total MET as well as phosphorylated AKT in primary human hepatocytes obtained from patients with 0%, 15% or 40% steatosis. To better stress that all experiments were performed with primary murine and human hepatocytes we now through-out the text included “primary” and to present our new results with the primary human hepatocytes adapted the text (344-356):

To test this hypothesis, we first examined in primary hepatocytes of three patients with 0%, 15% or 40% steatosis the levels of basal phosphorylated MET and total MET by quantitative immunoblotting (Fig. EV5). The quantification of the results showed that in relation to the levels of total MET there was indeed a significant increase in phosphorylated MET in the highly steatotic patient hepatocytes (Fig. EV5A), strikingly resembling the findings in the WD primary hepatocytes. Likewise, the phosphorylation of AKT after HGF stimulation was significantly reduced in the highly steatotic primary hepatocytes (Figure EV5B) suggesting that the effects we uncovered in the preclinical mouse model also occurred in patients. To further dissect the underlying mechanism and the relation to clinical outcome, we analyzed HGF-induced signal transduction in patient-derived primary human hepatocytes isolated from tumor-free tissue of seven patients with different liver pathologies that underwent partial liver hepatectomy (see Dataset EV5 for patient anamnesis).

2. The involvement of mTOR pathway in regulating AKT phosphorylation in WD hepatocytes is highlighted in the study, but only in the mathematical models. Experimental evidence from primary hepatocytes must be presented to validate the assumptions/conclusions.

To address the point raised by the reviewer, we utilized our global proteome data of the primary hepatocytes and analyzed the differential regulation of proteins associated with mTORC1 signal transduction. We included the results in the new Figure EV3C and adapted the text accordingly (line 259-267):

The development of the mathematical model pointed to an important role of the mTOR pathway in regulating AKT phosphorylation in WD hepatocytes. Therefore, we utilized our global proteome data of SD and WD hepatocytes and tested whether hallmark genes encoding proteins of mTORC1 signal transduction as listed in GSEA are differentially regulated in WD hepatocytes compared to SD hepatocytes. Of the 200 genes listed, 129 proteins were detected

across all samples. The expression values were scaled per protein and compared between primary hepatocytes from 9 WD mice and 9 SD mice. As shown in EV3C, the clustering of the data showed that based on the hallmarks of mTORC1 signal transduction, SD and WD mice separate from each other, confirming our observation.

3. It would be appropriate to show that ribosomal protein S6 phosphorylation is altered in conditions under evaluation using western blotting etc.

The analysis of S6 phosphorylation is included in Fig. 3 and we have now highlighted the difference in abundance in Fig 3b. This is now better pointed out in the text (line 231-234 and 248-252):

These determinations revealed a significant decrease in the intensity of MET, SIN1 and S6 and a significant increase of TSC in the WD primary hepatocytes, while the intensity of the other protein species was comparable between SD and WD primary hepatocytes, respectively (Fig 3B).

As a result, the final mathematical model, which included only one diet-specific dynamic parameter, the basal MET phosphorylation rate, and 11 diet-specific parameters for protein abundance, was able to explain the increased basal phosphorylation and reduced maximal induction of pMET, pERK and pS6 as well as the reduced phosphorylation of AKT in response to HGF stimulation in WD hepatocytes (Fig 3D).

4. Authors show that there is a marked increase in cell cycle entries and proliferation rate in WD compared to SD hepatocytes. Evidence must be presented to support that pMET levels in fact drive (or contribute significantly to) this. Current experimental data does not show this.

We agree that the evidence is indirect, but since we are working with steatotic primary murine and human hepatocytes, which are very sensitive and only loosely adhere to tissue culture plates, perturbations are barely possible. To strengthen our evidence, we now included in the new Figure 5D the results of a proliferation assay based on the quantification of the DNA content in HGF or unstimulated WD and SD primary hepatocytes, for which we observed the elevated basal phosphorylation of MET upon exposure to the WD. We have worked with Fucci mice for several years (e.g. Mueller et al, Mol Syst Biol 2015, DOI: 10.15252/msb.20156032) and so far did not come across differences in the dynamics of HGF induced signal transduction including MET phosphorylation and HGF induced proliferative responses compared to wild type primary hepatocytes. Furthermore, we now included as new Fig 5A and B the immunoblotting results of the analysis of basal total MET and basal phosphorylated MET in primary human hepatocytes derived from patients with elevated steatosis levels, which confirmed that also in patients an increase in steatosis correlates with a relative increase in the basal phosphorylated MET levels. We adapted the text accordingly (line 330-335) and (line 344-356):

To corroborate these findings, we utilized the fluorescent dye SYBR Green, which binds to double stranded DNA, to quantify the DNA content in SD and WD primary hepatocytes left unstimulated or stimulated with HGF. In agreement with the results obtained with the Fucci mouse derived primary hepatocytes, WD primary hepatocytes showed an increase in DNA content even in the absence of HGF, indicating that these primary hepatocytes were able to proliferate independently of HGF stimulation (Fig 5D).

To test this hypothesis, we first examined in primary hepatocytes of three patients with 0%, 15% or 40% steatosis the levels of basal phosphorylated MET and total MET by quantitative immunoblotting (Fig. EV5). The quantification of the results showed that in relation to the levels

of total MET there was indeed a significant increase in phosphorylated MET in the highly steatotic patient hepatocytes (Fig. EV5A), strikingly resembling the findings in the WD primary hepatocytes. Likewise, the phosphorylation of AKT after HGF stimulation was significantly reduced in the highly steatotic primary hepatocytes (Figure EV5B) suggesting that the effects we uncovered in the preclinical mouse model also occurred in patients. To further dissect the underlying mechanism and the relation to clinical outcome, we analyzed HGF-induced signal transduction in patient-derived primary human hepatocytes isolated from tumor-free tissue of seven patients with different liver pathologies that underwent partial liver hepatectomy (see Dataset EV5 for patient anamnesis).

5. Authors have used quantitative western blotting to estimate parameters used in the study, but none of the blots are presented in the paper/made available as supplementary. Immunoblot-based quantifications of the pMET level in the patient-derived hepatocytes are not presented either. All western blotting images must be made available. Also, catalog numbers of antibodies, etc. must be provided to enable reproducibility.

We entirely agree with the reviewer and included now original immunoblots in Figures EV3 and EV5. Furthermore, we are making all immunoblot images available and updated our material list.

6. Recovery experiments in WD hepatocytes by perturbing levels of pMET, phospho-AKT, mTOR components etc. would be appropriate for substantiating authors' claims about regulatory mechanisms of AKT phosphorylation and hepatocyte proliferation in WD/SD.

It would indeed be nice to directly perturb the levels of pMET, pAKT and mTOR components and examine the consequences. However, all our studies were performed with primary hepatocytes and in particular the steatotic hepatocytes are very challenging to obtain and cultivate as they only loosely adhere to cell culture plates. They would most likely not withstand the rather harsh conditions of siRNA knockdown experiments or treatment with inhibitors solubilized in DMSO. We therefore opted for validating our key observation, the impact of steatosis on phosphorylated MET and the corresponding reduction of pAKT, in primary human hepatocytes from patients with elevated steatosis levels. We would like to point out that these patient samples are very precious since patients with high degree of steatosis are infrequently operated. We could only obtain these samples through our close collaborations with Thomas Berg and Daniel Seehofer at the University Hospital in Leipzig, who are now included as new coauthors. We included the results as new Fig 5A and B and adapted the text accordingly (line 344-356):

To test this hypothesis, we first examined in primary hepatocytes of three patients with 0%, 15% or 40% steatosis the levels of basal phosphorylated MET and total MET by quantitative immunoblotting (Fig. EV5). The quantification of the results showed that in relation to the levels of total MET there was indeed a significant increase in phosphorylated MET in the highly steatotic patient hepatocytes (Fig. EV5A), strikingly resembling the findings in the WD primary hepatocytes. Likewise, the phosphorylation of AKT after HGF stimulation was significantly reduced in the highly steatotic primary hepatocytes (Figure EV5B) suggesting that the effects we uncovered in the preclinical mouse model also occurred in patients. To further dissect the underlying mechanism and the relation to clinical outcome, we analyzed HGF-induced signal transduction in patient-derived primary human hepatocytes isolated from tumor-free tissue of seven patients with different liver pathologies that underwent partial liver hepatectomy (see Dataset EV5 for patient anamnesis).

Minor comments:

7. Does Fig 3B represent an average of N=9 mice? If so, plot data from each mouse, as individual data points overlayed on the current averaged box plot.

We improved Fig 3B as suggested.

8. In discussion, line 431, change "proliferation of hepatocytes" to "ex-vivo proliferation of primary hepatocytes". The authors must highlight that observations made in the study are ex-vivo and not in-vivo.

We corrected this statement as suggested.

9. Line 330-332 repeat immediately after that.

We corrected this mistake.

10. Fig. EV3A shows the immunoblot used for generating a standard curve using recombinant AKT. The western blots used for AKT concentration assessment should also be included.

We improved Fig EV3A as suggested.

30th Nov 2023

Manuscript Number: MSB-2023-11856R

Title: Basal MET Phosphorylation is an Indicator of Hepatocyte Dysregulation in Liver Disease

Dear Ursula,

Thank you for sending us your revised manuscript. We have now heard back from the two reviewers who agreed to evaluate your revised study. As you will see below, the reviewers are satisfied with the performed revisions and support publication. As such, I am glad to inform you that we can soon accept your manuscript for publication, pending some minor revisions listed below, all related to editorial issues.

- As the nomenclature changed in June 2023, Nonalcoholic fatty liver disease (NAFLD) needs to be replaced by metabolic dysfunction-associated steatotic liver disease (MASLD) throughout the manuscript.
- Our data editors have indicated that the following needs to be corrected/added in the figure legends:
 - i) Please indicate the statistical test used for data analysis in the legends of figures 1e; 6b
 - ii) Please note that the box plots need to be defined in terms of minima, maxima, bounds of box and whiskers in the legends of figures 1a; 3b; 5d; EV3b; EV6c.
 - iii) Please note that information related to n is missing in the legends of figures 1e; EV3b; EV5a-b; EV6c.
 - iv) Please note that the error bars are not defined in the legends of figures 2b-c; 3d; 6a; EV1a-b; EV5a-b; EV6b
- Please make sure that the information provided in the manuscript text and in the submission system match. Currently, some of the funding information is missing from the submission system.
- Datasets EV1-EV7: please provide one file per EV Dataset. Please include the description of each EV Dataset in the dataset file itself, i.e. in a separate tab for .xls files or as a README.txt file in .zip folders for .csv files.
- The Source Data should be uploaded as individual files (one folder per figure).
- The synopsis image should be provided as a .jpg or .png file (exactly 550 px width). Please make sure that all labels are readable at this final size.

Please resubmit your revised manuscript online, with a covering letter listing amendments and responses to each point raised by the referees. Please resubmit the paper ****within one month**** and ideally as soon as possible. If we do not receive the revised manuscript within this time period, the file might be closed and any subsequent resubmission would be treated as a new manuscript. Please use the Manuscript Number (above) in all correspondence.

Click on the link below to submit your revised paper.

Best wishes,

Maria

Maria Polychronidou, PhD
Senior Editor
Molecular Systems Biology

If you do choose to resubmit, please click on the link below to submit the revision online before 30th Dec 2023.

IMPORTANT: Please note that corresponding authors are required to supply an ORCID ID for their name upon submission of a revised manuscript (EMBO Press signed a joint statement to encourage ORCID adoption).

(<https://www.embopress.org/page/journal/17444292/authorguide#editorialprocess>)

Currently, our records indicate that the ORCID for your account is 0000-0001-9845-3099.

Link Not Available

*** PLEASE NOTE *** As part of the EMBO Press transparent editorial process initiative (see our Editorial at <https://dx.doi.org/10.1038/msb.2010.72> , Molecular Systems Biology will publish online a Review Process File to accompany accepted manuscripts. When preparing your letter of response, please be aware that in the event of acceptance, your cover letter/point-by-point document will be included as part of this File, which will be available to the scientific community. More information about this initiative is available in our Instructions to Authors. If you have any questions about this initiative, please contact the editorial office (msb@embo.org).

Reviewer #1:

The authors have adequately adressed all of the points I raised during review.

Reviewer #3:

The authors have satisfactorily addressed most of my concerns

All editorial and formatting issues were resolved by the authors.

8th Dec 2023

Manuscript number: MSB-2023-11856RR

Title: Basal MET Phosphorylation is an Indicator of Hepatocyte Dysregulation in Liver Disease

Dear Ursula,

Thank you again for sending us your revised manuscript. We are now satisfied with the modifications made and I am pleased to inform you that your paper has been accepted for publication.

Kind regards,

Maria

Maria Polychronidou, PhD
Senior Editor
Molecular Systems Biology
